# Distinct dynamics of social motivation drive differential social behavior in laboratory rat and mouse strains

Shai Netser[1,5], Ana Meyer [2,5], Hen Magalnik [1], Asaph Zylbertal [3], Shani Haskal de la Zerda [1], Mayan Briller [1], Alexander Bizer [4], Valery Grinevich [2,6] & Shlomo Wagner [1,6✉]

Mice and rats are widely used to explore mechanisms of mammalian social behavior in health and disease, raising the question whether they actually differ in their social behavior. Here we address this question by directly comparing social investigation behavior between two mouse and rat strains used most frequently for behavioral studies and as models of neuropathological conditions: C57BL/6 J mice and Sprague Dawley (SD) rats. Employing novel experimental systems for behavioral analysis of both subjects and stimuli during the social preference test, we reveal marked differences in behavioral dynamics between the strains, suggesting stronger and faster induction of social motivation in SD rats. These different behavioral patterns, which correlate with distinctive c-Fos expression in social motivation-related brain areas, are modified by competition with non-social rewarding stimuli, in a strain-specific manner. Thus, these two strains differ in their social behavior, which should be taken into consideration when selecting an appropriate model organism.

[1] Sagol Department of Neurobiology, the Integrated Brain and Behavior Research Center (IBBRC), University of Haifa, Haifa 3498838, Israel. [2] Department of Neuropeptide Research in Psychiatry, Central Institute of Mental Health, University of Heidelberg, Mannheim J5, 69159, Germany. [3] Department of Neuroscience, Physiology and Pharmacology, University College London, London WCE1 6BT, UK. [4] Faculty of Natural Sciences, University of Haifa, Haifa 3498838, Israel. [5] These authors contributed equally: Shai Netser, Ana Meyer. [6] These authors jointly supervised this work: Valery Grinevich, Shlomo Wagner. ✉email: shlomow@research.haifa.ac.il

Animal models are valuable tools for studying the biological mechanisms underlying mammalian social behavior in general, and particularly human pathologies associated with atypical social behavior[1]. Yet, mammalian social interactions are extremely complex, involving affiliative or aggressive behaviors toward specific individuals. Moreover, human social relationships are highly dynamic, emotional, and experience-based. Thus, using animal models for studying social behavior requires methodologies that consider and monitor the dynamics of social behavior.

Rats and mice have been leading model organisms in behavioral and biomedical research for well over a century[2]. In recent decades, the availability of a much larger genetic toolbox for mice, particularly embryonic stem-cell-based targeting technology for gene modification, has made mice the major model of choice in biomedical research[3–5]. Nevertheless, with the recent emergence of tools for altering the rat genome, notably genome-editing technologies such as CRISPR/Cas9, the technological gap between the two organisms is closing, and an ever-growing number of genetically modified rat lines becomes available for scientists[6–8]. It is therefore becoming vital to consider the physiological, anatomical, biochemical, and behavioral differences between the various laboratory strains of rats and mice when choosing the most appropriate model system for a specific biological question[9,10]. Moreover, although the existence of significant differences in social behavior between rats and mice is widely recognized[2,11], only few previous studies have directly compared their social behavior and supplied quantitative analysis of such differences.

Previously, we presented an automated experimental system enabling a detailed analysis of social investigation behavior in small rodents[12,13]. Here we use this system to compare the temporal pattern (henceforth termed dynamics) of social behavior between the two rodent strains most frequently used for behavioral studies and as a genetic background for most models of neuropathological conditions: C57BL/6J mice and Sprague-Dawley (SD) rats. We describe marked differences in social behavior between these two strains. Specifically, SD rats show immediate and strong motivation to interact with novel social stimuli, while C57BL/6J mice exhibit a low level of such motivation. Unlike C57BL/6J mice, the immediate high social motivation of SD rats persists during a competition between social and food stimuli following food deprivation. Moreover, we find that the stimulus's movements attract SD rat subjects while deterring C57BL/6J mice subjects. Following characterization of these behavioral distinctions, we further employ c-Fos staining and computational modeling to demonstrate that the behavioral differences between SD rats and C57BL/6J mice reflect the distinct dynamics of their social motivation.

## Results

### Social preference differs between C57BL/6J mice and SD rats.

We used identical experimental setups, adapted to the different size of the animals[12], to conduct the social-preference and social-novelty preference (SP/SNP) tests as previously described by us[13], with both C57BL/6J mice and SD rats.

Figure 1a–d summarizes the results of the SP test, performed by C56BL/6J adult male mice ($n = 58$, Supplementary Video 1). Throughout the test, the subjects investigated the social stimulus more than they investigated the object, thus displaying a clear social preference (Fig. 1a). We have also analyzed the transitions the subject mice made between the two stimuli along the time course of the test (Fig. 1b). We found a high level of transitions at early stages, with a clear peak of the mean transition rate (red line) around 50 s from the test start, and a reduction in this

parameter to a stable low rate (~50% of peak value) later during the test. Interestingly, these changes in transition rate correlated in time with changes in the duration of investigation bouts, categorized by us according to their duration (see full distributions and explanation in Supplementary Fig. 1) into three groups: short (<6 s), intermediate (6–19 s), and long (>19 s). As apparent in Fig. 1c, d, short bouts of investigation, which may reflect curiosity per se, showed no difference between the stimuli, and decreased with time (2-way repeated ANOVA, stimuli—$F_{1,57} = 1.098$, $p = 0.290$; time—$F_{3,171} = 16.777$, $p < 0.0001$; stimuli × time—$F_{3,171} = 0.525$, $p = 0.665$). In contrast, long bouts, which showed significant differences between the two stimuli throughout the test and seemed to reflect interactions between the subject and stimuli, increased in their total time during the test (stimuli—$F_{1,57} = 24.323$, $p < 0.0001$; time—$F_{3,171} = 3.212$, $p = 0.029$; stimuli × time—$F_{3,171} = 0.168$, $p = 0.917$). Accordingly, we previously suggested[13] that in C57BL/6J mice, the SP test may be divided into two behavioral phases: an early exploratory phase, characterized by numerous transitions between stimuli and short investigation bouts, and a late interactive phase, characterized by a low transition rate and extended investigation bouts.

Figure 1e–h summarizes the results of adult male SD rats ($n = 60$) using the same SP/SNP paradigm as used with C57BL/6J mice. SD rats showed a stronger preference than mice for the social stimulus, compared to object, throughout the SP test (Fig. 1e), and these differences in social preference were statistically significant (see below Fig. 2e). Even more strikingly and in sharp contrast to C57BL/6J mice, SD rats almost did not perform transitions between stimuli at early stages of the SP test (Fig. 1f). Rather, in almost all cases, they started the test with intensive investigation of the social stimulus, and only later on began showing interest in the object (Supplementary Video 2). The enhanced interest of SD rats in the social stimulus at early stages of the test, was also reflected by the dynamics of short and long bouts. The time dedicated by SD rats for short investigation bouts (Fig. 1g), despite being significantly different between stimuli, did not change (stimuli—$F_{1,59} = 1000.373$, $p < 0.0001$; time—$F_{3,177} = 1.259$, $p = 0.289$; stimuli × time—$F_{3,177} = 2.117$, $p = 0.099$). In contrast, the time dedicated for long bouts (Fig. 1h), which were largely observed only for the social stimulus (see Fig. 2a, d), peaked during the first minute of the test (stimuli × time—$F_{3,177} = 12.003$, $p < 0.0001$). Thus, male SD rats showed stronger social preference than C57BL/6J mice and unlike them, intensively interacted with the social stimulus already at the very beginning of the test. Similar differences were found by us between females of the same two strains (Supplementary Fig. 2), proving that these differences are not sex-specific. To make sure that the initial low level of social preference and interactions in C57BL/6J mice is not due to a higher level of baseline anxiety, we used the 20-min habituation period preceding the SP test as an open-field test. We calculate the center/periphery location ratio as a measure for anxiety: the higher the ratio, the lower the anxiety it reflects. We found that C57BL/6J mice exhibited higher center/periphery ratio than SD rats (mean ± SD: mice—$0.22 \pm 0.22$, rats—$0.08 \pm 0.05$; $t$ test, $t = 3.251$, d$f = 71$, $p < 0.001$), thus excluding the possibility that a higher baseline level of anxiety in C57BL/6J mice makes them less social than SD rats. Notably, these results are in accordance with a recent study using both open-field and elevated-plus maze tests to examine anxiety levels in C57BL/6J mice and SD rats[14].

### Variations in SP behavior are strain-specific.

A direct comparison between the performance of C57BL/6J mice and SD rats in the SP test is displayed in Fig. 2. The marked difference in behavioral dynamics between the two strains is clearly visible

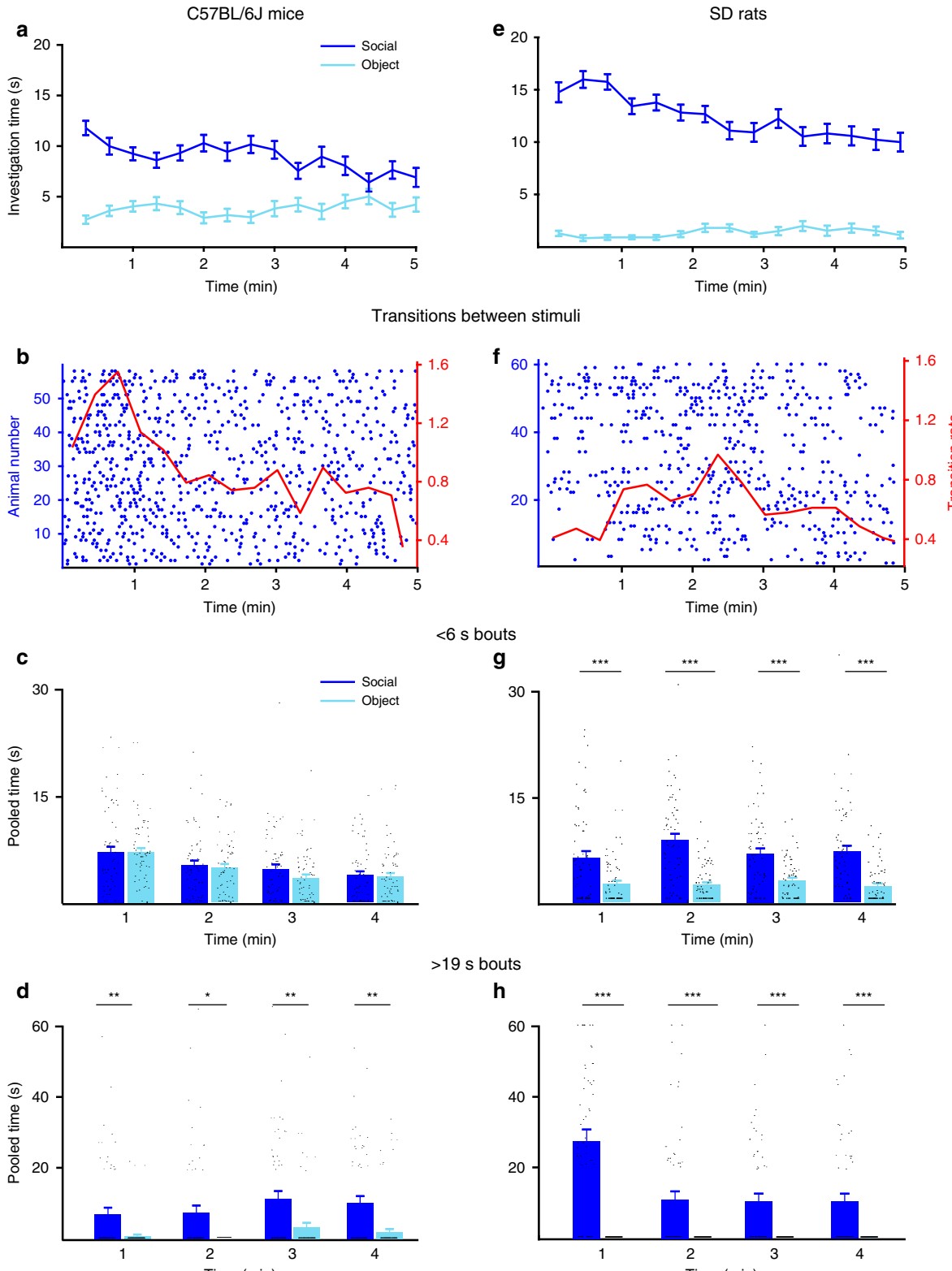

**Fig. 1 Distinct social-preference (SP) dynamics of C57BL/6J mice and SD rats. a** Mean investigation time of social and object stimuli, averaged in 20-s bins across time during the 5-min-long SP test conducted with C57BL/6J mice ($n = 58$). **b** Transitions between the two stimuli, made by subject mice across time during the test. Each punctum denotes the beginning of investigation of a new stimulus, and each row represents a single subject. The mean rate (using 20-s bins) is denoted by the red line (right red y-axis). **c** Mean pooled time of short investigation bouts (<6 s) across time during the SP test of mice (using 1-min bins, last minute excluded; see "Methods"). **d** As in **c**, when extended bouts (>19 s) are considered. Black lines at the bottom of the bars represent data points with a value of zero. **e**–**h** As in **a**–**d**, for SD rats ($n = 60$). Black lines at the bottom of the bars represent data points with a value of zero. *$p < 0.05$, **$p < 0.01$, ***$p < 0.001$, post hoc two-tail $t$ test following the main effect. All error bars represent SEM. Source data are provided as a Source Data file.

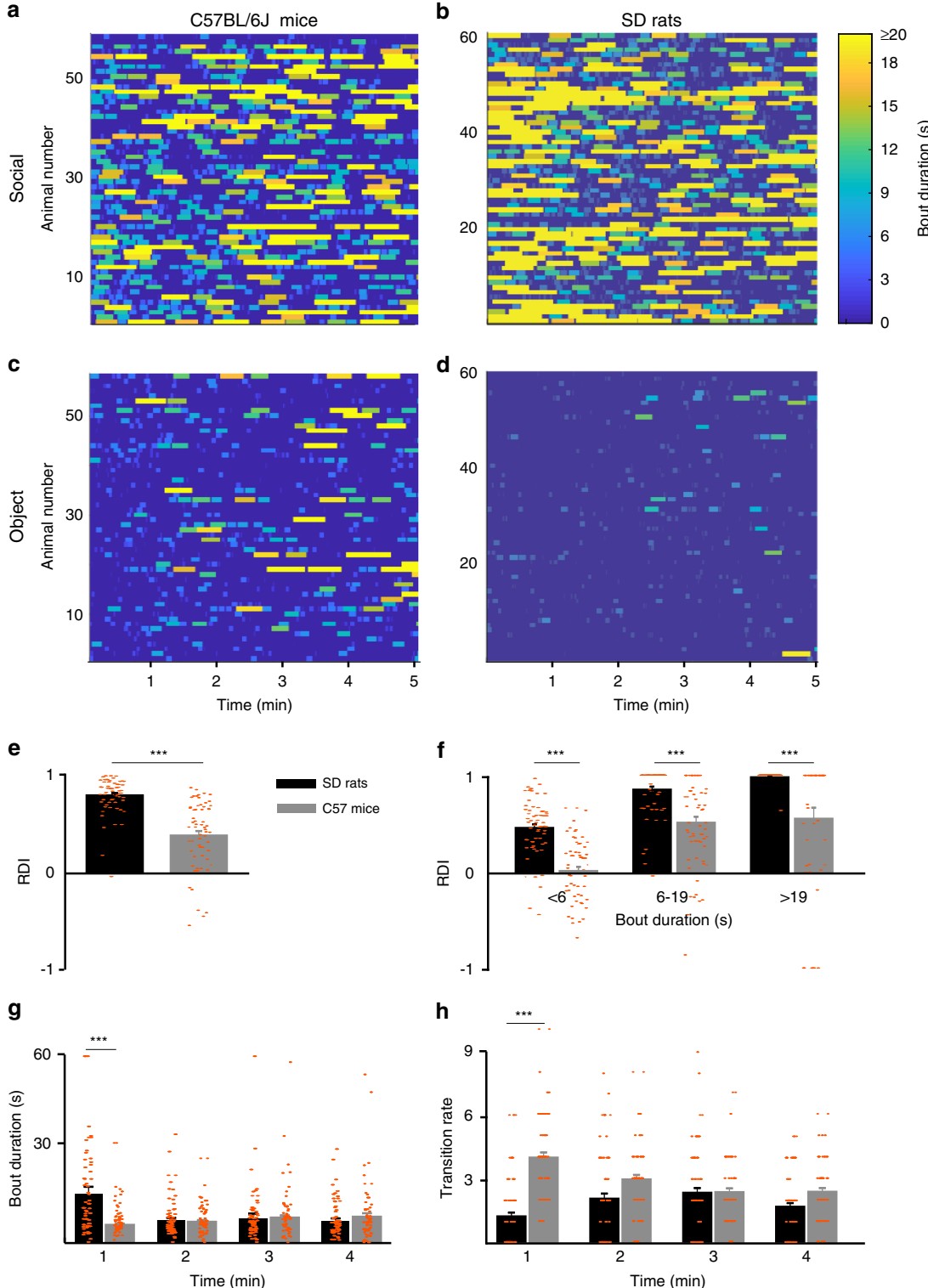

**Fig. 2 Differences in social-preference dynamics are at the beginning of the test. a** Heatmap (each row represents one animal) of the investigation behavior toward the social stimuli during the SP test, performed by C57BL/6J mice (*n* = 58). **b** As in **a** for SD rats (*n* = 60). **c, d** As in **a, b** for object stimuli. **e** Mean RDI (relative differential investigation) values for the SP test conducted with C57BL/6J mice and SD rats. **f** As in **e**, for the three categories of bout duration. **g** Comparison of mean bout duration values (averaged using 1-min bins) for the first 4 min of the test, between C57BL/6J mice and SD rats. **h** As in **g**, for mean transition rate. ***\*p* < 0.001, post hoc two-tail *t* test following the main effect. All error bars represent SEM. Source data are provided as a Source Data file.

from the heatmaps of their investigation behavior (Fig. 2a–d). The relative differential investigation of the two stimuli (RDI) shows statistically significant higher values for SD rats (Fig. 2e), suggesting much higher preference for the social stimulus, as compared to C57BL/6J mice ($t$ test, $t = 23.619$, df $= 114$, $p < 0.0001$). This difference in RDI was statistically significant in all three categories of bout duration (Fig. 2f; strain × time—$F_{2,144} = 361.26$, $p < 0.0001$). Yet, when the mean bout duration was compared between SD rats and C57BL/6J mice across time (Fig. 2g), we found a significantly longer bout duration for SD rats only in the first minute of the test (strain—$F_{1,116} = 0.435$, $p = 0.039$; time—$F_{3,348} = 240.73$, $p < 0.0001$; strain × time—$F_{3,348} = 0.206$, $p = 0.65$). Similar results (with opposite relationship) were obtained by comparing the mean transition rate between SD rats and C57BL/6J mice across time (Fig. 2h). We found a significantly higher rate of transitions for C57BL/6J mice as compared to SD rats, specifically during the first minute of the test (strain × time—$F_{3,348} = 6.704$, $p = 0.010$). Thus, it seems as if the difference between C57BL/6J mice and SD rats in the SP test is most significant at the early phase of the test.

So far, we found marked differences in SP behavior between C57BL/6J mice (inbred strain) and SD rats (outbred strain), the two main laboratory strains of rats and mice. Yet, these differences may be specific to these two strains, or may be common to other strains of rats and mice, thus reflecting species-specific differences. To explore this question, we conducted SP experiments with adult male ICR (CD-1) mice and Wistar Hannover rats, both of which are outbred strains. As apparent in Fig. 3a–h, both these strains showed a pattern of SP dynamics that looks more similar to SD rats than to C57BL/6J mice. This was reflected by the relatively high preference for the social stimulus over the object (Fig. 3a, b), relatively low transition rate at the beginning of the test (Fig. 3c, d), lack of clear reduction in short bouts along the test time course (Fig. 3e, f; two-way repeated ANOVA, mice: stimuli—$F_{1,22} = 0.516$, $p = 0.480$; time—$F_{3,66} = 0.256$, $p = 0.857$; stimuli × time—$F_{3,66} = 0.475$, $p = 0.701$; rats: stimuli—$F_{1,19} = 5.923$, $p = 0.025$; time—$F_{3,57} = 0.503$, $p = 0.682$; stimuli × time—$F_{3,57} = 0.381$, $p = 0.767$), and the highest level of long bouts at the first minute of the test (Fig. 3g, h; mice: stimuli—$F_{1,22} = 29.221$, $p < 0.0001$; time—$F_{3,66} = 1.204$, $p = 0.315$; stimuli × time—$F_{3,66} = 0.980$, $p = 0.407$; rats: stimuli—$F_{1,19} = 30.31$, $p < 0.0001$; time—$F_{3,57} = 1.197$, $p = 0.319$; stimuli × time—$F_{3,57} = 1.149$, $p = 0.337$). A statistical analysis, however, revealed that these two strains are located in between SD rats and C57Bl/6J mice, as reflected by the RDI (Fig. 3i, Welch's ANOVA, $F_{3,57.745} = 26.112$, $p < 0.001$), the first-minute transition rate (Fig. 3j, Welch's ANOVA, $F_{3,58} = 22.071$, $p < 0.001$), as well as by the first-minute bout duration (Fig. 3k, Welch's ANOVA, $F_{3,49.872} = 4.106$, $p = 0.011$). In order to make sure that these differences are not due to the fact that C57BL/6J mice are an inbred strain, as opposed to the other strains, we also examined the behavior of BALB/c mice, another inbred laboratory mouse strain. We found that these mice showed a behavioral pattern that is almost identical to ICR mice, but significantly different from C57BL/6J mice (Supplementary Fig. 3). Thus, it seems as if the differences observed by us in the behavioral dynamics during the SP test are strain-specific rather than species-specific, with C57BL/6J mice and SD rats representing two extremes. We therefore focused the rest of the study on these two strains.

**No strain-specific differences in SNP.** Analysis of the behavior of male C57BL/6J mice in the SNP test revealed identical dynamics as in the SP test preceding it (Fig. 4a–d). Here, too, we observed an initial high rate of transitions followed by a gradual reduction with time (Fig. 4b). As in the SP test, short bouts

(Fig. 4c) did not differ between stimuli and diminished during the test (stimuli—$F_{1,57} = 0.468$, $p = 0.496$; time—$F_{3,171} = 16.917$, $p < 0.0001$; stimuli × time—$F_{3,171} = 1.061$, $p = 0.3$), while long bouts (Fig. 4d) significantly differed between stimuli and increased along the time course of the test (stimuli—$F_{1,57} = 12.347$, $p = 0.0008$; time—$F_{3,171} = 4.870$, $p = 0.002$; stimuli × time—$F = 0.360$, $p = 0.78$). Thus, C57BL/6J mice showed similar behavioral dynamics in both the SP and SNP tests, characterized by a tendency for stimuli exploration at the beginning of the test, and by increasing interaction with stimuli, mainly with the preferred one, at later stages.

Surprisingly, when SD rats performed the SNP test following a 5-min-long SP test as in mice (Supplementary Fig. 4A, short paradigm), we observed no preference for the novel social stimulus (Supplementary Fig. 4B, C, $t$ test, $t = 1.923$, df $= 9$, $p = 0.086$). Interestingly, when we conducted the same test with ICR mice and Wistar Hannover rats, we found that ICR mice behaved like SD rats, with no apparent SNP (Supplementary Fig. 4D, E), while Wistar Hannover rats behaved like C57BL/6J mice and exhibited clear preference of the novel conspecific following a 5-min-long SP test (Supplementary Fig. 4F, G). These results further support the conclusions that the differences in social behavior observed by us in the SP and SNP tests are strain-specific rather than species-specific, and that these differences are not related to the question of whether the examined strain is inbred or outbred.

We therefore extended the exposure of the subject to the familiar social stimulus by conducting the SP test for 15 min (Supplementary Fig. 4A, extended paradigm), in order to obtain a clear preference of SD rats for the novel social stimulus in the SNP test that followed (Supplementary Fig. 4H, I, $t$ test, $t = 3.996$, df $= 19$, $p < 0.0001$). Interestingly, the dynamics of SNP behavior displayed by SD rats were rather similar, qualitatively and quantitatively, to those of C57BL/6J mice (see Fig. 4e–h), with a high level of transitions at the beginning of the test, followed by a gradual reduction later on (Fig. 4f). Moreover, as in mice, the short bouts (Fig. 4g) did not differ between stimuli and diminished with time (stimuli—$F_{1,58} = 0.637$, $p = 0.427$; time—$F_{3,174} = 11.724$, $p = p < 0.0001$; stimuli × time—$F_{3,174} = 0.153$, $p = 0.92$). A deviation from the results obtained using C57BL/6J mice was evident only in the long bouts (Fig. 4h), which as in mice were mainly observed toward the novel social stimulus, but in contrast to mice, did not significantly increase in time (stimuli—$F_{1,58} = 12.626$, $p < 0.0001$; time—$F_{3,174} = 1.889$, $p = 0.133$; stimuli × time—$F_{3,174} = 0.611$, $p = 0.608$). Thus, while C57BL/6J mice demonstrated the same behavioral dynamics in both the SP and SNP tests, SD rats showed highly distinct dynamics between these two tests.

In order to find which of these cases is the exception, we conducted another test, which is similar to the SP and SNP tests while using a distinct set of social stimuli—the sex-preference test (SxP). In this test, the subjects are simultaneously exposed to same- and opposite-sex stimuli in the same experimental system and conditions used for the SP and SNP tests. We found that male C57BL/6J mice ($n = 45$) and SD rats ($n = 20$), both of which clearly preferred female over male stimuli, showed similar behavioral dynamics, which were comparable to the dynamics observed in the SNP test (Supplementary Fig. 5). Thus, it seems as if the differences between SD rats and C57BL/6J mice are revealed only in the SP test, most probably because in this test, the subject needs to choose between a social stimulus and an object, rather than two social stimuli as in the SNP and SxP tests.

**Differences in social preference stem from the subjects.** The differences in behavioral dynamics between C57BL/6J mice and

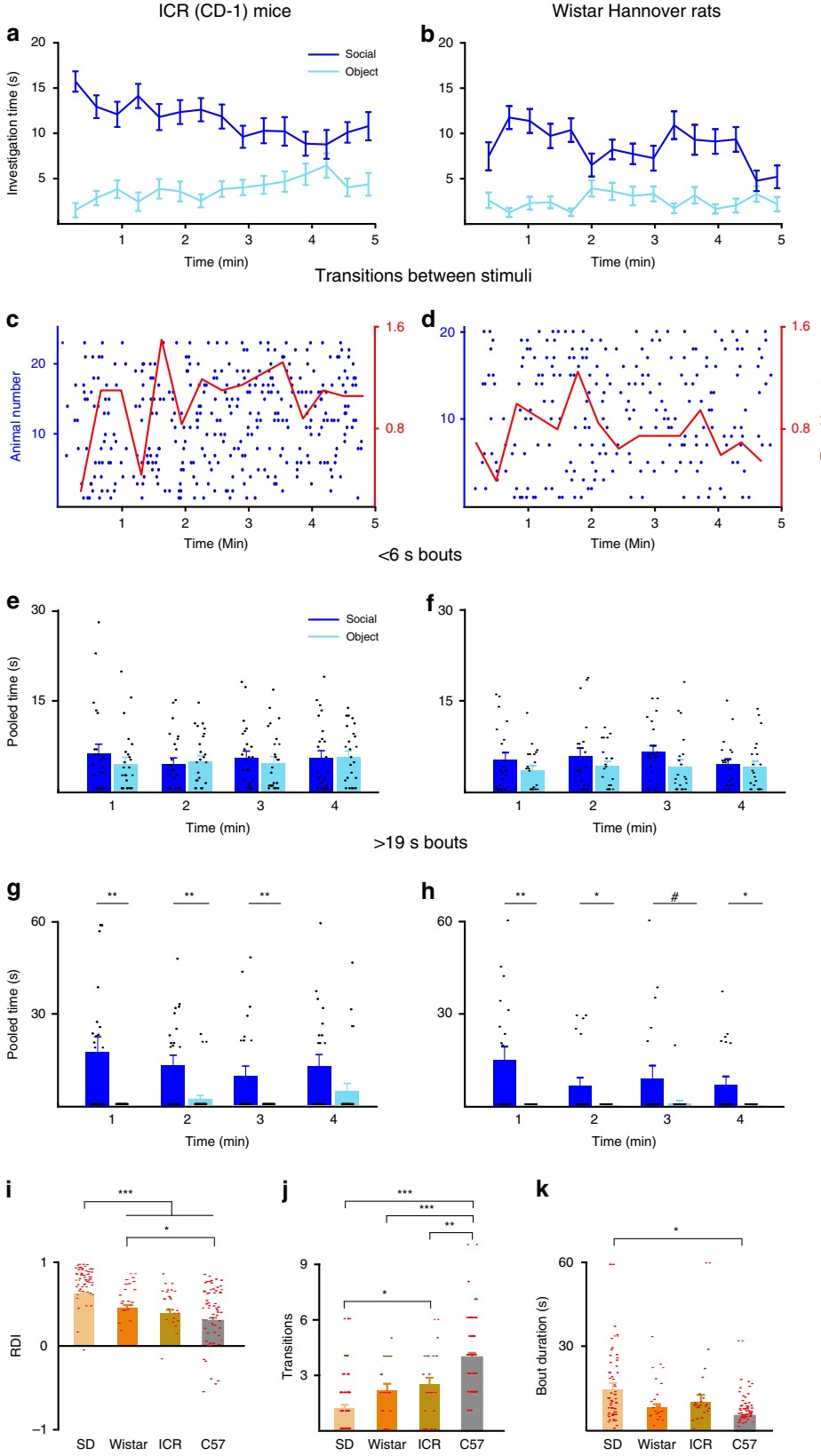

SD rats in the SP test may be due to distinctions in behavior of the social stimuli rather than the subjects themselves. To the best of our knowledge, the behavior of the social stimuli in similar tests has never been reported. To assess this behavior, we constructed a movement-monitoring system, comprising an array of piezoelectric sensors located at the floor of the triangular chamber containing the social stimulus (Fig. 5a, b). We then recorded the electrical signals generated by the sensors, which reflect the movement of the social stimulus, during the SP test. The raw signals recorded along the time course from C57BL/6J mice

**Fig. 3 Differences in social-preference dynamics are strain-specific. a** Mean investigation time of social and object stimuli, averaged in 20-s bins for ICR (CD-1) mice ($n = 23$) across time during the 5-min-long SP test. **b** As in **a**, for Wistar Hannover rats ($n = 20$). **c** Transitions between the two stimuli, made by subject ICR (CD-1) mice across time during the test. Each punctum denotes the beginning of investigation of a new stimulus, and each row represents a single subject. The mean rate (using 20-s bins) is denoted by the red line (right red y-axis). **d** As in **c**, for Wistar Hannover rats. **e** Mean pooled time of short investigation bouts (<6 s) made by ICR (CD-1) mice across time during the SP test (using 1-min bins, last minute excluded; see "Methods"). **f** As in **e**, for Wistar Hannover rats. **g** As in **e**, when extended bouts (>19 s) are considered. *$p < 0.05$, **$p < 0.01$, post hoc two-tail $t$ test following the main effect. Black lines at the bottom of the bars represent data points with a value of zero. **h** As in **g**, for Wistar Hannover rats. #$p = 0.07$, *$p < 0.05$, **$p < 0.01$, post hoc two-tail $t$ test following the main effect. Black lines at the bottom of the bars represent data points with a value of zero. **i** Mean RDI values for the four strains of rats and mice tested with the SP test (C57BL/6J: $n = 58$, SD rats: $n = 60$, ICR: $n = 23$, and Wistar Hannover: $n = 20$). ***$p < 0.001$, *$p < 0.05$, post hoc two-tail Games–Howell test following the main effect. **j** Mean transition rate during the first minute of the SP test, for the four strains of rats and mice tested, as in **i**. *$p < 0.05$, **$p < 0.01$, ***$p < 0.001$, post hoc two-tail Games–Howell test following the main effect. **k** Mean bout duration during the first minute of the SP test, for the four strains of rats and mice tested, as in **i**. *$p < 0.05$, post hoc two-tail Games–Howell test following the main effect. All error bars represent SEM. Source data are provided as a Source Data file.

stimuli ($n = 28$) showed silent periods with no movement, periods of minor movements, and several brief events of major movements reflected by sharp peaks of the recorded signal (Fig. 5c). Following normalization of the raw signals, we defined a major movement as an event generating a signal that crossed ±20% of the maximal absolute value. When the number of major movements was calculated, C57BL/6J mice showed a rather constant mean value across the time course of the experiments (Fig. 5d; one-way repeated ANOVA, $F_{4,112} = 0.724$, $p = 0.57$).

We then asked whether the subjects adapted their investigation behavior according to the movement of the social stimulus during the SP test. As shown in Fig. 5e, where the mean social investigation time of the subject (orange and red traces) is superimposed on the mean movement of the stimulus (blue trace), the tendency of C57BL/6J mice for social investigation following major movements of the stimulus (red trace) was lower than their tendency to do so without any large movement by the stimulus (orange trace). In contrast, no such difference was observed when object investigation was measured (Fig. 5f). The difference in investigation time of the social stimulus following its movement, as opposed to no movement of the stimulus, was statistically significant (Fig. 5g; two-way repeated ANOVA, stimuli × movement—$F_{1,27} = 4.26$, $p = 0.04$). These results suggest that C57BL/6J mouse subjects avoid investigating the social stimulus immediately following a major movement made by it.

We then analyzed the signals obtained with SD rats ($n = 24$) performing similar experiments (Fig. 5h), for which we found a gradual reduction in the major movements performed by the social stimuli along time (Fig. 5i, one-way repeated ANOVA, $F_{4,92} = 4.009$, $p = 0.004$). In contrast to C57BL/6J mice, SD rats exhibited increased tendency to explore the social stimulus (Fig. 5j), and decreased tendency to explore the object (Fig. 5k), immediately following a major movement of the social stimulus (Fig. 5l; two-way repeated ANOVA, stimuli × movement—$F_{1,23} = 7.008$, $p = 0.014$). Thus, both C57BL/6J mice and SD rats seem to adjust their investigation behavior to the movements of the social stimulus. However, while SD rat subjects seem to be attracted by movements of the social stimulus, C57BL/6J mouse subjects seem to be deterred by them. We concluded that the marked differences between these two strains in the SP test are mainly due to distinctions between the subjects, rather than the stimuli.

To make sure that the distinctions in social preference between SD rats and C57BL/6J mice are not limited to the condition of restricted interactions between subjects and stimuli, dictated by our experimental system, we compared free male–male interactions of adult subjects and juvenile social stimuli between C57BL/6J mice ($n = 12$) and SD rats ($n = 14$). We found that even in this condition, SD rats showed significantly higher interaction time than C57BL/6J mice (Supplementary Fig. 6A, $t$ test, $t = 3.909$,

df $= 24$, $p = 0.001$), mainly due to significantly higher level of long (>6 s), but not short (<6 s) interactions between SD rats, compared to C57BL/6J mice (Supplementary Fig. 6B; mixed-model ANOVA, time × strain—$F_{1,24} = 6.444$, $p = 0.018$).

**Strain-specific c-Fos induction by a novel social stimulus**. We then aimed to identify neural substrates that may underlie the behavioral differences we observed between SD rats and C57BL/6J mice. As demonstrated above (Fig. 2), these differences were most significant during the first minute of the SP test. We therefore analyzed the expression of the neuronal activation marker c-Fos, induced during the early phase of the SP test in several social behavior-associated brain areas of C57BL/6J mice and SD rats[15–17]. To that end, group-housed animals were taken directly from their home cage to the experimental arena for a 15-min habituation period. Following this period, the subjects were exposed to either two empty chambers (control group, four mice and four rats), two chambers with a social stimulus in one of them (social chamber group, six mice and five rats), or a social stimulus freely moving in the arena (free-interaction group, four mice and four rats). After 2 min of exposure, the stimuli were removed, and the subjects were left in the arena for 90 min to allow for c-Fos expression, and then sacrificed. We reasoned that since the animals were free to interact with their cagemates in their home cage until 15 min prior to the exposure to stimuli, any significant induction of c-Fos expression in the brain following the social encounter, as compared to control animals, ought to be elicited merely by the exposure to the novel social stimulus. As apparent from the heatmaps depicted in Fig. 6a, b for the social chamber group, during the 2-min test, most C57BL/6J mice behaved as expected, displaying relatively weak social preference, short investigation bouts, and many transitions (Fig. 6c). In contrast, SD rats exhibited a strong preference for the social stimulus and extended investigation bouts toward it (Fig. 6d, e). Moreover, they almost did not conduct any transition between stimuli during this period (Fig. 6f). Analysis of c-Fos staining was initially conducted in four brain areas associated with social investigation[18]: medial amygdala (MeA), nucleus accumbens (NAc), and the dorsal (LSD) and ventral (LSV) lateral septum, for each of the three groups (control, social chamber, and free interaction) (Fig. 6g, h). A statistical analysis did not reveal any significant difference between the three groups of C57BL/6J mice, for any of the examined brain regions (Fig. 6i; one-way ANOVA, MeA—$F_{2,11} = 0.84$, $p = 0.457$; NAc—$F_{2,11} = 1.88$, $p = 0.197$; LSD—$F_{2,11} = 1.68$, $p = 0.229$; LSV—$F_{2,11} = 1.26$, $p = 0.319$). In contrast, SD rats exhibited brain region-specific differential expression of c-Fos (Fig. 6j). As apparent, both the MeA and NAc, but not LSD and LSV, showed significantly higher levels of c-Fos in rats exposed to novel social stimuli, either through a chamber or during free interactions, as

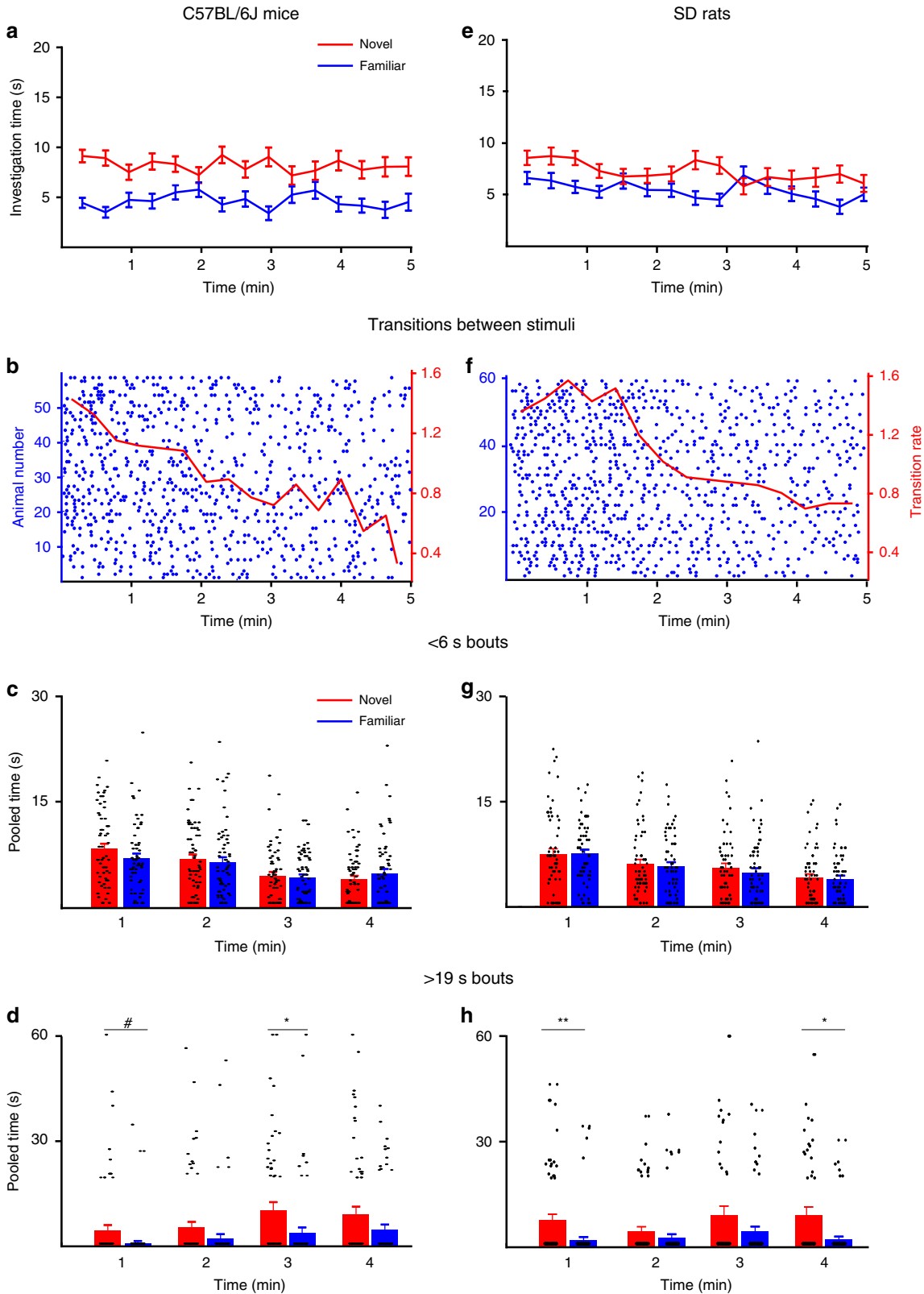

compared to the control group (one-way ANOVA, MeA—$F_{2,10} = 13.46$, $p = 0.0015$; NAc—$F_{2,10} = 5.62$, $p = 0.0231$; LSD—$F_{2,10} = 3.43$, $p = 0.073$; LSV—$F_{2,10} = 1.21$, $p = 0.336$). Similar results were found in later analysis of the ventral tegmental area (VTA) of the same animals (Supplementary Fig. 7). Since both the NAc and VTA play a major role in motivational behavior in

general[19,20], and particularly in the motivation for social interaction[21–25], these results suggest the involvement of social motivation in the distinct dynamics of social behavior of C57BL/6J mice and SD rats.

As our behavioral experiments showed that the tendency of C57BL/6J mice for social interaction increases with time during

**Fig. 4 Similar social-novelty preference (SNP) dynamics of C57BL/6J mice and SD rats. a** Mean investigation time of novel and familiar social stimuli, averaged in 20-s bins across time during the 5-min-long SNP test conducted with C57BL/6J mice ($n = 58$). **b** Transitions between the two stimuli, made by subject mice across time during the test. Each punctum denotes the beginning of investigation of a new stimulus, and each row represents a single subject. The mean rate (using 20-s bins) is denoted by the red line (right red y-axis). **c** Mean pooled time of short investigation bouts (<6 s) across time during the SNP test of mice (using 1-min bins, last minute excluded; see "Methods"). **d** As in **c**, when extended bouts (>19 s) are considered. Black lines at the bottom of the bars represent data points with a value of zero. **e–h** As in **a–d**, for SD rats ($n = 60$ rats). #$p = 0.05$, *$p < 0.05$, **$p < 0.01$, post hoc two-tail $t$ test following the main effect. All error bars represent SEM. Source data are provided as a Source Data file.

the SP test, we predicted that if given enough time of exposure to social stimuli, C57BL/6J mice will also show c-Fos induction in the MeA and NAc. To challenge this prediction, we conducted a similar experiment as described above, with a new cohort of C57BL/6J mice that were tested with either 2- or 5-min-long SP tests, and compared their c-Fos expression to control animals that were exposed to objects only ($n = 5$ animals/group). This time, we separately analyzed the NAc shell and core, since these regions were recently reported to play differential roles in emotional behavior[26]. As in the previous experiment, we found no significant induction of c-Fos expression in any of the examined murine brain regions following a 2-min SP test. After a 5-min SP test, however, we observed a significant (or almost significant) increase of c-Fos induction specifically in the MeA and NAc shell, but not in the LSD, LSV, or NAc core (Supplementary Fig. 8; one-way ANOVA, LSV—$F_{2,12} = 0.847$, $p = 0.453$; LSD—$F_{2,12} = 2.327$, $p = 0.140$; NAc core—$F_{2,12} = 0.444$, $p = 0.652$; NAc shell—$F_{2,12} = 3.767$, $p = 0.057$; MeA—$F_{2,12} = 7.161$, $p = 0.009$). Overall, these results suggest that SD rats and C57BL/6J mice differ in the dynamics of c-Fos induction, hence of neuronal activation, during a social encounter, specifically in brain regions associated with social motivation.

**Differential social motivation revealed by competing stimuli.** Following the results described so far, we hypothesized that an encounter of a SD rat with a novel social stimulus elicits a high level of motivation for social interaction from its very beginning, which afterward gradually decreases with time, most probably due to the reduction in stimulus novelty. In contrast, a C57BL/6J mouse starts an encounter with a novel social stimulus displaying a low level of social motivation, reflected by a high level of transitions and a low level of social interactions, which afterward may gradually increase with time (see model in Fig. 8o below).

To challenge this hypothesis, we developed a behavioral paradigm, conducted in the behavioral experimental system described above. In this paradigm, we induced a competition between a novel social stimulus and another rewarding stimulus, while gradually increasing the rewarding value of the nonsocial stimulus. To that end, we used food pallets located in the chamber opposite to the social stimulus in a manner that prevents the subject from consuming them (Fig. 7a). We gradually increased the rewarding value of the food palettes by preventing food from the subjects for increasing periods (Fig. 7b). We then analyzed the behavior of C57BL/6J mice ($n = 16$) and SD rats ($n = 8$) in these conditions, in a similar manner to the SP test. For both mice and rats, we found a statistically significant interaction between starvation time and preference (Fig. 7c, d; two-way repeated ANOVA, stimuli × time, mice—$F_{2,30} = 11.332$, $p < 0.0001$; rats—$F_{2,14} = 117.507$, $p < 0.0001$). We then analyzed the preference for the distinct starvation times and found that even in satiety state, C57BL/6J mice did not prefer the social stimulus over the food (paired $t$ test following the main effect, $t = -1.454$, d$f = 15$, $p = 0.167$). Moreover, they exhibited clear and significant preference for the food over the social stimulus already following 4 h of food deprivation ($t = 3.351$, d$f = 15$, $p = 0.004$) and this preference got even stronger following 24 h of starvation ($t = 4.118$,

d$f = 15$, $p = 0.0001$) (Fig. 7c). In contrast to C57BL/6J mice, SD rats did show a strong social preference at satiety ($t = -8.448$, d$f = 7$, $p < 0.001$), and even following 24 h of starvation, they did not prefer the food over a social stimulus ($t = 1.799$, d$f = 7$, $p = 0.115$). Nevertheless, after 48 h of starvation, SD rats did show a clear and significant preference for food over the social stimulus ($t = 4.331$, d$f = 7$, $p = 0.003$) (Fig. 7d). Assuming similar influence of food starvation on the motivation to eat (see "Discussion"), the differences between C57BL/6J mice and SD rats clearly demonstrate a much higher motivation in SD rats to interact with a novel social stimulus, as compared to C57BL/6J mice.

In order to further analyze the dynamics of social motivation in C57BL/6J mice and SD rats, we focused on the specific condition when no preference to any of the stimuli was displayed. We reasoned that the equal motivation to investigate the competing stimuli in this condition would best expose dynamic changes in social motivation. Indeed, C57BL/6J mice at satiety, the condition when they exhibited no general preference, started the test with a clear bias toward the food, which was lost after about a minute (Fig. 7e). In sharp contrast, SD rats at 24 h of starvation, while showing no general preference, started the test with a clear preference for the social stimulus over the food, and only after about a minute lost this preference and investigate both stimuli to the same extent (Fig. 7f). Statistical analysis of social investigation time in these conditions revealed a significant interaction between time and strain (Fig. 7g; mixed-model ANOVA, time × strain—$F_{4,88} = 5.805$, $p < 0.0001$). Post hoc analysis showed a significant difference in social investigation between SD rats and C57BL/6J mice only at the first minute of the test ($t$ test following the main effect, $t = -4.842$, d$f = 22$, $p < 0.001$). Notably, when comparing the mean transition rate during the first minute between the various periods of food deprivation, we found a gradual reduction in transition rate with longer periods of food deprivation for C57BL/6J mice (Fig. 7h), and an opposite tendency in SD rats (Fig. 7i). The differences in transition rate between satiety and the longest period of food deprivation (24 h for mice, 48 h for rats), were statistically significant (Fig. 7h, i; Wilcoxon signed-rank test, mice—$p = 0.001$, rats—$p = 0.034$). Thus, by changing the motivation balance of the animals between food and social stimuli, we managed to induce distinct behavioral dynamics in the same task using the same stimuli, and these changes were opposite between SD rats and C57BL/6J mice.

Overall, the above results suggest that C57BL/6J mice and SD rats exhibit marked differences in the dynamics of their motivation to interact with a novel social stimulus, and that these motivational differences drive their distinct patterns of social investigation during the SP test.

**Computational model supports the role of social motivation.** Finally, to further test our hypothesis regarding the role of social motivation in the behavioral differences between C57BL/6J mice and SD rats, we constructed a simplified computational discrete-time Markov model of social behavior during the SP and SNP tests. This model (Fig. 8a, b, see Table 1) comprises four behavioral states: stimulus 1 investigation (S1), stimulus 2 investigation (S2), stillness (S3), and arena exploration (S4). The state of

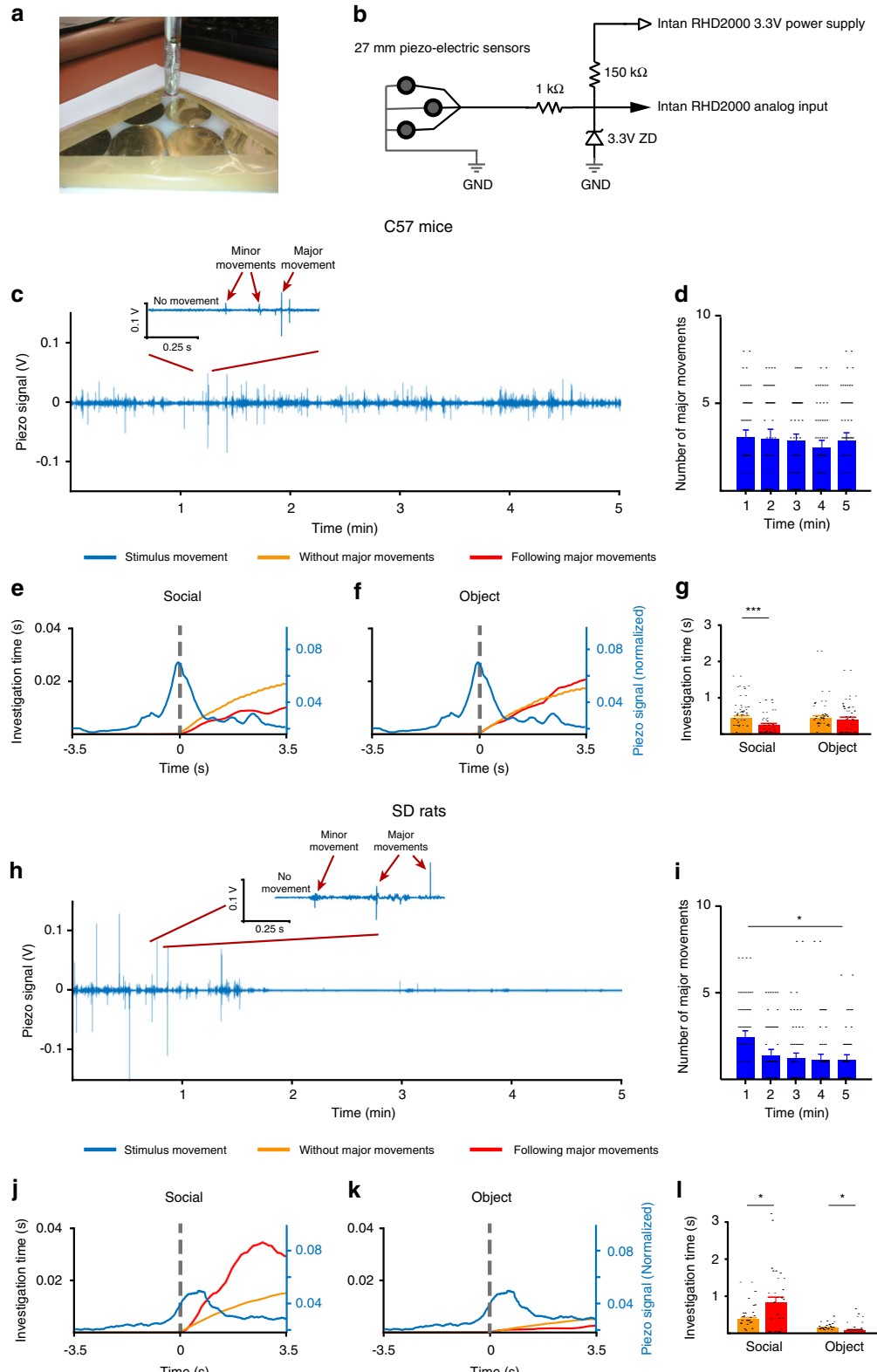

stillness represents all cases when the animal is not exhibiting motivational behavior, i.e., not moving. For simplification, we assumed that every motivational state (*S1*, *S2*, and *S4*) has to end at stillness (*S3*), before a new motivational state may arise. Thus, there was no possible direct transition between *S1*, *S2*, and *S4*. The dynamic transition probabilities of the model depend on three

dynamic variables: stimulus 1 and stimulus 2 reward values and the anxiety level. In our model, anxiety was defined as a product of two variables: the stress level, assumed by us to be directly related to the novel environment, thus continually decreasing during the test due to habituation, and the difference in reward between the two stimuli. In other words, the smaller the

**Fig. 5 Different responses of SD rat and C57BL/6J mouse subjects to movements made by social stimuli. a** Array of piezoelectric sensor disks spread on the floor of a triangular chamber. **b** A scheme of the electric circuit used for acquiring data from the piezoelectric sensors during experiments. **c** An example of a raw-signal trace recorded during a 5-min SP test from a mouse social stimulus. The inset shows a short section in higher time resolution. **d** Mean number of large movements (averaged using 1-min bins) performed by mouse social stimuli (n = 28) across the time course of the SP test. **e** Development of social investigation following a major stimulus movement (red trace) or without major movement (orange trace), superimposed on the mean signal of the stimulus movement (blue trace, right y-axis). Dashed line represents the peak of the mean movement. **f** Same as in **e**, for investigation of the object stimulus during the same experiments. **g** Mean cumulative investigation time of the social (left) or object (right) stimuli, with (red) or without (orange) social stimulus movement. **h-l** As in **d-g**, for rats (n = 24). *p < 0.05, ***p < 0.001, post hoc two-tail t test following the main effect. All error bars represent SEM. Source data are provided as a Source Data file.

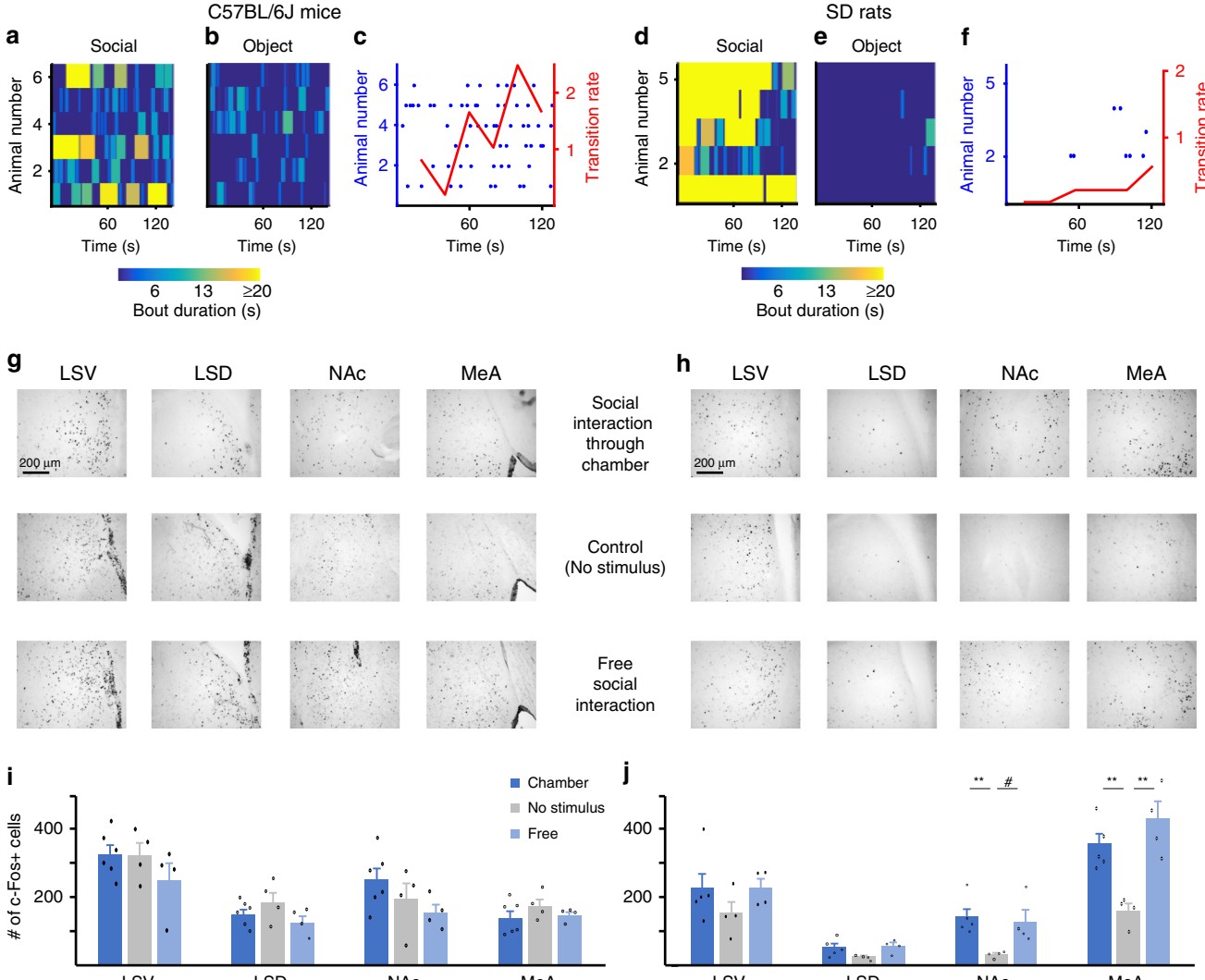

**Fig. 6 C57BL/6J mice and SD rats differ in c-Fos induction by exposure to novel social stimuli. a** Heatmap of investigation behavior of six C57BL/6J mice toward the social stimuli in a 2-min SP test, with each line representing a single animal. **b** As in **a**, for the object stimuli. **c** Analysis of transitions between stimuli made during the same experiments described in **a**, **b**, with each transition represented by a punctum and each animal by a row. Mean values (averaged over 20-s bins) are shown by the red line (right y-axis). **d-f** As in **a-c**, for five SD rats. **g** Representative images of c-Fos immunostaining in the four brain areas denoted above, for one mouse from each of the three groups. **h** Same as **g**, for one rat from each of the three groups (the groups are denoted between **g** and **h**). **i** Mean values of the three groups of mice (Control: n = 4, Chamber: n = 6, Free: n = 4), for each of the four brain regions. **j** Same as **i**, for rats (Control: n = 4, Chamber: n = 5, Free: n = 4). #p < 0.07, **p < 0.01, post hoc two-tail t test with Bonferroni's correction following the main effect. All error bars represent SEM. Source data are provided as a Source Data file.

difference in reward between the stimuli, the higher the anxiety level of the subject, due to the difficulty of choosing between the two competing stimuli, which creates a motivational conflict[27,28]. In this model, the only difference between C57BL/6J mice and SD rats was the dynamics of social reward, with SD rats showing an initial high level of social reward, which gradually decreased

during the test, and C57BL/6J mice showing an initial low level, which gradually increased. We used an evolutionary multi-objective optimization algorithm to fit the model parameters using the experimental results of the SP test in C57BL/6J mice (see Table 2). Running the model with these parameters (Fig. 8c) on a virtual sample of 60 C57BL/6J mice resulted in dynamics of

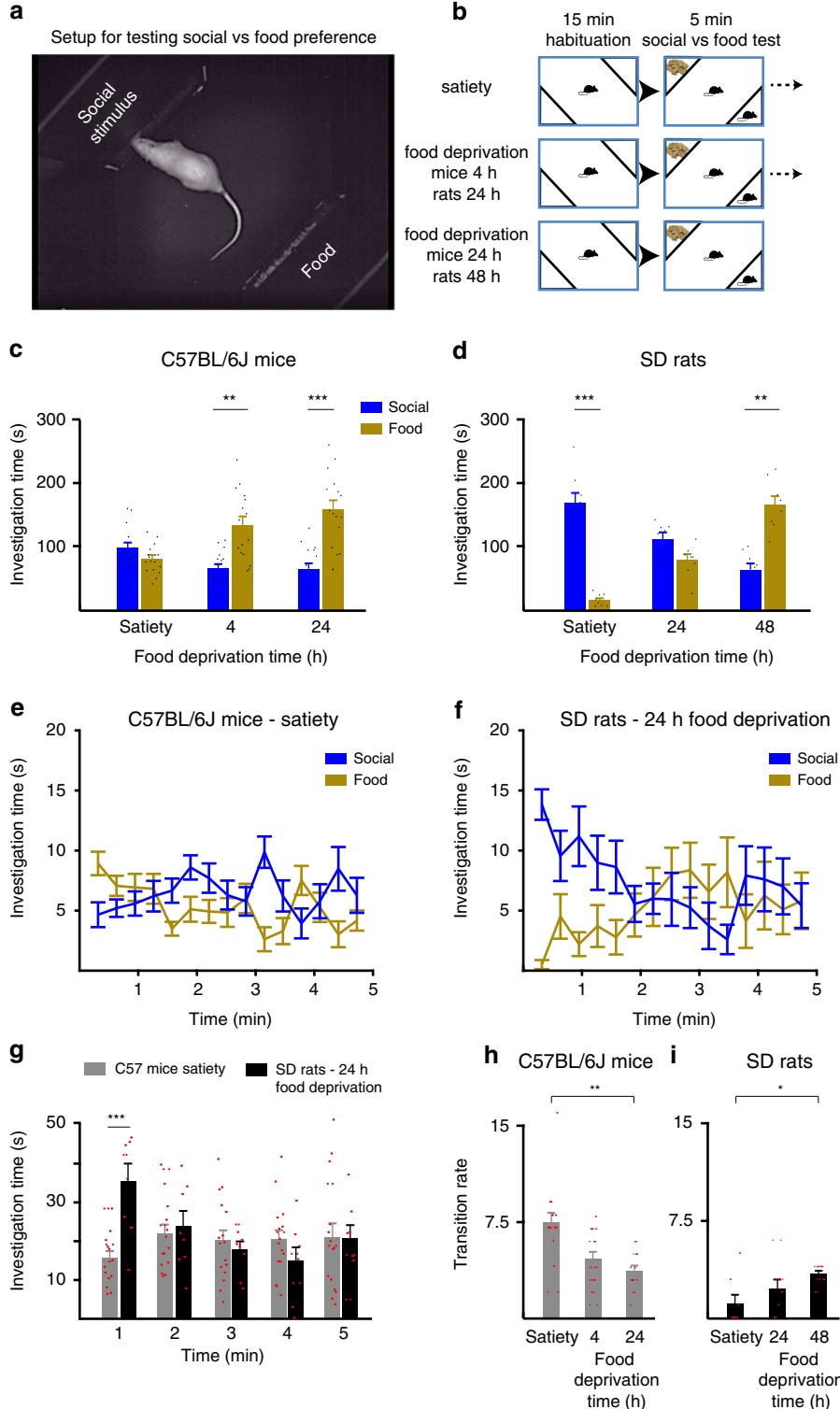

**Fig. 7 Differences in social motivation of C57BL/6J mice and SD rats, as revealed by competition between social and food stimuli. a** A picture depicting the setup used for creating a competition between social and food stimuli, without allowing the subject to consume the food palettes. **b** Schematic representation of the experimental timeline, in which the same groups of C57BL/6J mice ($n = 16$) and SD rats ($n = 8$) were tested following three different conditions of food deprivation, as denoted. **c** Mean total investigation time (during the 5-min test) of social and food stimuli by mouse subjects, at the three different conditions of food deprivation. **d** As in **c**, for rats. **e** Mean investigation time (20-s bins) of social and food stimuli by mice subjects across time during the test, for satiety condition. **f** Same as **e**, for rats following 24 h of food deprivation. **g** Statistical comparison of mean social investigation time (1-min bins) across time during the test, between SD rats and C57BL/6J mice at conditions were no general preference was exhibited. **h** Mean transition rate for the first minute of the test for the experiments shown in (**c**). **i** As in **h**, for the experiments shown in (**d**). *$p < 0.05$, **$p < 0.01$, ***$p < 0.001$, post hoc two-tail $t$ test following the main effect. All error bars represent SEM. Source data are provided as a Source Data file.

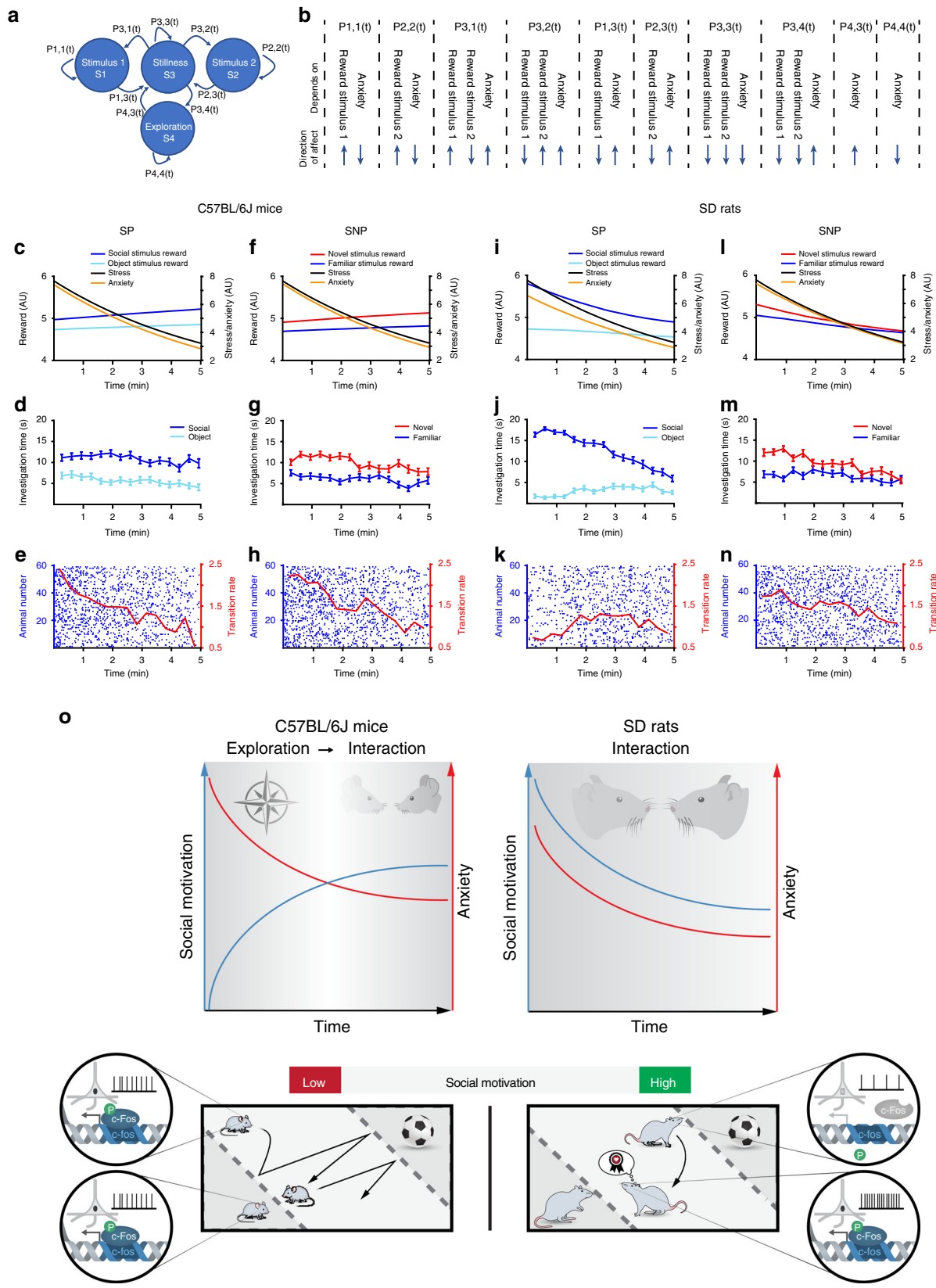

social behavior (Fig. 8d) and transitions between stimuli (Fig. 8e) that were highly similar to the experimentally observed results (Fig. 1a, b). The distributions of bout length in the model were also similar to the experimental results (Supplementary Fig. 9).

Then, we modified the initial values of social reward in the model to fit those expected for the SNP test, i.e., a similar value of the novel stimulus as in the SP test, and a lower value for the familiar one (Fig. 8f) and run it again on a sample of 60 mice. Despite not

**Fig. 8 Behavioral differences between SD rats and C57BL/6J mice are recapitulated by a computational model. a** A schematic depiction of a Markov model with four behavioral states used to simulate the behavior of SD rats and C57BL/6J mice in the SP and SNP tests. **b** A schematic description of how each of the model's probabilities is affected by the various parameters. **c** The four parameters of the model plotted against time for the SP test in mice. **d** Mean investigation time (averaged over 20-s bins) for the social and object stimuli, of a simulated random population of 60 mice in the SP test. **e** Transitions between stimuli made by the same simulated experiment as in **d**, with each punctum representing a transition and each row a subject mouse. Mean transition rate (averaged over 20-s bins) is represented by the red line (note the right y-axis for the red line). **f–h** As in **c–e**, for the SNP test simulated with mice. **i–k** As in **c–e**, for the SP test simulated for a random population of 60 rats. **l–n** As in **c–e**, for the SNP test simulated with rats. **o** A cartoon, schematically describing the opposite dynamics of social motivation between SD rats and C57BL/6J mice (upper panels), leading to the distinct patterns of social behavior of the two strains in the SP test. These patterns are described by a gradual shift from exploration to interaction in mice, as opposed to a shift from intensive to more relaxed social interactions in rats. They lead to a high level of transitions between stimuli in mice, as opposed to rats (lower panel), which is explained by rather similar motivation to explore both stimuli in mice, in contrast to the much higher motivation to explore the social stimulus exhibited by rats. These levels of motivation are reflected by the rate of neuronal activity in social-motivation-associated brain regions, which leads to c-Fos transcription in the nuclei of these neurons. All error bars represent SEM. Source data are provided as a Source Data file.

**Table 1 The transition probabilities between the various states of the model.**

|  | State 1 | State 2 | State 3 | State 4 |
|---|---|---|---|---|
| State 1 | $(e^{-(\beta 1_r r_1 + \beta 1_a a)} + 1)^{-1}$ | 0 | $1 - (e^{-(\beta 1_r r_1 + \beta 1_a a)} + 1)^{-1}$ | 0 |
| State 2 | 0 | $(e^{-(\beta 1_r r_2 + \beta 1_a a)} + 1)^{-1}$ | $1 - (e^{-(\beta 1_r r_2 + \beta 1_a a)} + 1)^{-1}$ | 0 |
| State 3 | $\dfrac{e^{\beta 2_r r_1}}{e^{\beta 2_r r_1} + e^{\beta 2_r r_2} + e^{S_0 \beta 2_a a} + e^{E_0}}$ | $\dfrac{e^{\beta 2_r r_2}}{e^{\beta 2_r r_1} + e^{\beta 2_r r_1} + e^{S_0 \beta 2_a a} + e^{E_0}}$ | $\dfrac{e^{S_0 \beta 2_a a}}{e^{\beta 2_r r_1} + e^{\beta 2_r r_1} + e^{S_0 \beta 2_a a} + e^{E_0}}$ | $\dfrac{e^{E_0}}{e^{\beta 2_r r_1} + e^{\beta 2_r r_1} + e^{S_0 \beta 2_a a} + e^{E_0}}$ |
| State 4 | 0 | 0 | $1 - (e^{-(S_0 + E_0 - \beta 2_a a)} + 1)^{-1}$ | $(e^{-(S_0 + E_0 - \beta 2_a a)} + 1)^{-1}$ |

**Table 2 Fixed model parameters and their values.**

| Parameter | Description | Mice SP | Mice SNP | Rats SP | Rats SNP |
|---|---|---|---|---|---|
| $\alpha_0$ | Initial anxiety | 7.66 | | | |
| $r_{10}$ | Initial reward for stimulus 1 | 4.98 | 4.7 | 5.8 | 5.05 |
| $r_{20}$ | Initial reward for stimulus 2 | 4.72 | 4.9 | 4.72 | 5.3 |
| $\tau_r$ | Reward - time constant | 100,406 | | 33,406 | |
| $\tau_a$ | Anxiety - time constant | 10,355.62 | | | |
| $\beta 1_r$ | The effect of reward on the decision to continue interacting with a stimulus | 1.1 | | | |
| $\beta 2_r$ | The effect of reward on the decision to choose a particular stimulus from stillness | 1.12 | | | |
| $\beta 1_a$ | The effect of anxiety on the decision to continue interacting with a stimulus | −0.08 | | | |
| $\beta 2_a$ | The effect of anxiety on the decision to continue staying in the exploration state | 0.71 | | | |
| $S_0$ | Stillness coefficient | 2.72 | | | |
| $E_0$ | Exploration coefficient | 5.19 | | | |

changing any other parameter from the initial optimization, the model results of the SNP test (Fig. 8g, h) were highly similar to those obtained experimentally (Supplementary Fig. 10). We then tested the same model for SD rats by modifying only the social reward values and dynamics, assuming a significantly higher initial reward value for social stimuli as compared to objects, and a gradual reduction in the reward values with time, with no additional changes (Fig. 8i, l). Surprisingly, the results of both SP and SNP simulation for the rat model were highly similar to the experimentally observed data (Fig. 8j, k, m, n, Supplementary Figs. 11 and 12). Notably, model rats displayed stronger social preference and opposite dynamics of transitions in the SP test, as compared to the SNP test and to model mice in both tests, exactly as observed experimentally. Thus, by using a simple model, fitted only to a single experimental paradigm in mice, we could recapitulate our observations throughout all paradigms in both C57BL/6J mice and SD rats, solely by modifying the dynamics of social reward according to our hypothesis. These results further support our hypothesis that the marked differences between C57BL/6J mice and SD rats in the dynamics of their investigation behavior in the SP test, are mainly due to differences in their social motivation systems, as schematically described in Fig. 8o.

## Discussion

In this study, we used the dynamics of social investigation behavior between the two rat and mouse strains most frequently used in social behavioral studies and as a genetic background for most models of human neuropathological conditions: C57BL/6J mice and SD rats. We found a major difference between SD rats and C57BL/6J mice in two aspects of their behavior. First, we observed weaker SNP in SD rats than in C57BL/6J mice (Fig. 4), that may be related to differences between these strains in social recognition memory, which were previously discussed in several review papers[29,30]. Second, we demonstrated a clear, well-defined difference between SD rats and C57BL/6J mice in the dynamics of their social investigation behavior in the SP test (Fig. 1). This difference between SD rats and C57BL/6J mice, which had been the focus of the study presented here, was linked by us to distinctions in their motivation to interact with a novel social stimulus.

There are substantial differences in the natural social structure of wild Norway rats (*Rattus norvegicus*) and house mice (*Mus musculus*)[2]. Although both species live in large hierarchical groups, rats are much less territorial and the hierarchy between males is far from absolute[31,32]. Accordingly, it is quite common

for all males in a group to mate with all females[33,34]. In contrast, mice live in more territorial structures founded by a single male that mates with multiple females[35]. Mouse burrows are thus much less complex than rat burrows, and usually only have a single cavity occupied by a single male[36]. As a result, interactions between male mice are less common and are more aggressive and territorial in nature than in rats[11]. Yet, the widely accepted conception that rats are generally more social and less aggressive in male–male interactions than mice[2] has rarely been examined in a direct, quantitative way. In 5-min-long dyadic interactions between conspecific strangers within an open-field arena of 1 m in diameter, *R. norvegicus* adult males made quicker and more numerous contacts with their partners and had a lower mean distance from them as compared to *M. musculus* males[37]. Thus, the results of this study suggest that rats show higher tendency for affiliative male–male interactions than mice. These results were confirmed and extended in laboratory animals by a recent study[38], which used the conditioned place preference (CPP) test to show that SD rats display higher tendency for social conditioning of place preference than C57BL/6J mice. Moreover, in animals that were concurrently conditioned for social interaction vs. cocaine, the relative reward strength for cocaine was 300-fold higher in C57BL/6J mice than in SD rats. These results suggest that male SD rats are more attracted by social interactions with other males than C57BL/6J mice, in accordance with our conclusions.

While the above-mentioned studies may suggest a species-specific difference in the motivation for male–male interactions between mice and rats, we found that CD-1 mice and Wistar Hannover rats, two more strains of laboratory animals, do not exhibit significant differences in their social investigation behavior during the SP test. Instead, the parameters characterizing their behavior were found to be located in between those of C57BL/6J mice and SD rats (Fig. 3i–k). Similar results were obtained from BALB/c mice, another inbred mouse strain (Supplementary Fig. 3). While these results cannot rule out the possibility of species-specific differences in social behavior between rats and mice, they suggest a spectrum of strain-dependent social investigation behavior, in which SD rats and C57BL/6J mice seem to represent two opposite edges. Our results showing that Wistar Hannover rats exhibit SNP after 5 min of exposure to a novel conspecific, similarly to C57BL/6J mice, while ICR mice behave like SD rats in this test, further support a strain-specific rather than species-specific difference, and suggest that it does not matter whether the strain is inbred or outbred. Overall, our results suggest that referring social behavior results from one strain to another, even within the same species, should be done cautiously.

We observed significant differences in the dynamics of social behavior during the SP test between C57BL/6J mice and SD rats (Fig. 2). These differences may be due to distinctions between the two strains in the dynamics of their motivation for social interaction that creates a stronger drive for male–male interactions in SD rats than in C57BL/6J mice at the beginning of the test. Yet, an alternative explanation is that the difference between the two strains stems from the behavior of the social stimuli, rather than the motivation of the subjects. The issue of stimulus behavior is a painful blind spot in all types of social recognition and discrimination tests, as no adequate analysis of stimulus behavior and the subject's response to it has been reported so far. Theoretically, the preference for one stimulus over the other may be driven by stimulus behavior as much as it may reflect recognition of the stimulus's unique blend of chemosensory cues by the subject[39]. Here, we directly addressed this issue by developing an experimental system, based on piezoelectric sensors that automatically report the movement of the stimulus in parallel to the

video recording of subject behavior (Fig. 5a, b). Using this system, we found that stimuli movements were not significantly different between SD rats and C57BL/6J mice, at least for the early phase of the test when most behavioral differences between these two strains were observed (Fig. 5d, i). Nevertheless, the reaction of the subjects to stimuli movements was markedly different between the two strains. Whereas C57BL/6J mice avoided investigating the social stimulus following a major movement of it, SD rats seemed to be attracted to the stimulus following major movements (Fig. 5g, l). These results demonstrate higher motivation of SD rats for interaction with novel social stimuli as compared to C57BL/6J mice.

To explore if these behavioral distinctions are also reflected by differences in the activity of a certain brain region, we examined c-Fos expression following 2-min-long SP tests and found that SD rats exhibited significant increase in c-Fos expression specifically in the MeA and NAc, while C57BL/6J mice did not (Fig. 6). These results seem contradictive to multiple studies reporting induction of c-Fos expression in various murine brain areas, including the MeA and NAc, following social interaction[40,41]. It should be noted, however, that in all these studies, the subjects were kept in social isolation for at least 1 week before the social encounter, a condition that increases their motivation for social interactions[42]. Moreover, even in these conditions, the NAc of male C57BL/6J mice was found to be activated only following an encounter with a female, but not with a male conspecific[41]. In contrast, our analysis of c-Fos induction by male–male interactions was performed using group-housed subjects tested 15 min after their transfer from their home cages to the experimental arena. In such conditions, one may expect a relatively high level of baseline (control) c-Fos expression due to interactions with cagemates that took place just before the test, while any increase in c-Fos expression following the SP test would reflect induction by the encounter with the novel social stimulus. Therefore, the brain region-specific significant increase in c-Fos expression observed by us in SD rats already following a 2-min social encounter was most likely induced by the exposure to a novel social stimulus. The relatively higher level of baseline c-Fos expression in C57BL/6J mice, observed by us in several brain regions, may be due to the high sensitivity of murine c-Fos expression to the novelty of the arena, reported by several previous studies[43,44]. Since our behavioral experiments with C57BL/6J mice showed a gradual increase in social interactions across the time course of the SP test, we predicted that if given enough time for this test, C57BL/6J mice would also exhibit brain induction of c-Fos expression. As predicted, in a separate experiment, we found again no c-Fos induction in the NAc and MeA of C57BL/6J mice following a 2-min SP test, but a significant induction following a 5-min test (Supplementary Fig. 8). It should be noted that similar results were previously reported for multiple brain areas using 1- and 3-min social encounters[45].

Several recent studies suggest that both the MeA and NAc are involved in rewarding social activities such as social play in rats[46–49] and mother–infant bonding in humans[50]. Notably, the MeA of male rats was shown to project to both the NAc and VTA[51], thus supplying a neuronal substrate for its involvement in regulating social motivation. It is worth mentioning that in a previous study, we found a strong induction of theta rhythmicity in both MeA and NAc of behaving SD rats during male–male social interactions, and this rhythmicity showed high coherence between these two brain areas[52].

The role of the NAc, a central part of the dopaminergic mesolimbic pathway, in mediating social reward and motivation, is well established in both rats and mice[22,25,53–55]. In their groundbreaking study, Gunaydin and colleagues[56] used fiber photometry in behaving mice to record the activity of

dopaminergic inputs arriving to the NAc from the VTA, strongly associated with reward in general, during social interactions. Interestingly, this activity (see Fig. 7 there) had a clear peak between 100 and 150 s after the beginning of social interaction, exactly the period when, in our experiments, C57BL/6J mice stopped exploring both stimuli and started interacting with the social stimulus. In contrast, a different study that used fast-scan cyclic voltammetry to record transient dopaminergic events in the NAc of behaving SD rats during social interactions[57], reported that these transients, which became sixfold more frequent during male–male interactions as compared to solitude, were the highest in rate at the very beginning of the interaction and dwindled significantly afterward. These studies are in perfect match with our results and further suggest distinct dynamics of social motivation between SD rats and C57BL/6J mice.

To directly compare the motivation for social interaction between SD rats and C57BL/6J mice, we developed a behavioral paradigm in which the animals are exposed to a novel social stimulus in one side of the arena and to food palettes in the other side (Fig. 7a). We conducted these experiments following various periods of food deprivation, thus increasing the rewarding values of food while keeping the social stimulus's rewarding value unchanged (Fig. 7b). We found that C57BL/6J mice exhibited no preference for social stimuli over food at satiation state, and showed clear food preference already after 4 h of food deprivation (Fig. 7c). In contrast, SD rats displayed strong social preference at satiation, and even after 24 h of food deprivation, did not prefer food over social interactions (Fig. 7d). Since both the percentage of reduction in body weight and the plasma insulin level were reported to be rather similar between SD rats and Swiss albino mice following 24 h of fasting[58], our results support a much higher motivation for social interactions in SD rats as compared to C57BL/6J mice. This conclusion is in accordance with the results of a previous study that compared social interactions with cocaine injection in a CPP test[38]. Moreover, when analyzing the dynamics of investigation behavior in the conditions when no general preference between food and social stimuli was observed, we found a significant difference between SD rats, which showed initial social preference, and C57BL/6J mice, which showed initial food preference (Fig. 7e–g). These results support the conclusion we drew from the SP test, that the strongest difference in social motivation between SD rats and C57BL/6J mice is at the very beginning of the test. Furthermore, when we examined the effect of food deprivation on the transition rate during the first minute of the test, a parameter clearly reflecting the different dynamics of social behavior between SD rats and C57BL/6J mice (Fig. 3j), we found that starvation gradually changes this parameter in both strains but in opposite directions (Fig. 7h, i). Thus, the changes in the balance of motivation to investigate the competing stimuli seem to drive changes in the dynamics of social behavior in both SD rats and C57BL/6J mice.

To further challenge this conclusion, we constructed a computational model of the behavior of SD rats and C57BL/6J mice in the SP and SNP tests (overall four tests, Figs. 1 and 4). This model, which to our best knowledge is the first to describe this type of behavior, is unrealistic and simplified, based on only four behavioral states and four emotional/motivational parameters: stress, anxiety, and the reward value of each of the stimuli (Fig. 8a). We defined stress as the deviation from homeostasis[59], caused in our case by the novel environment experienced by the subject, and assumed that it does not differ between the strains and gets gradually reduced across time due to habituation. Anxiety was defined as a more complex parameter[60] that in our case reflects both the stress level and the difference in reward between the two stimuli. We presumed that the need to choose between two stimuli with similar values will create a motivational

conflict and increase the anxiety level of the subject, in accordance with multiple studies associating motivational conflict and uncertainty with anxiety[27,28,61]. In contrast, a strong difference between the stimuli will solve the conflict and reduce the subject's anxiety. This may explain why SD rats show very different behavioral dynamics between the SP test, where they have much higher motivation to explore the social stimulus as compared to the object, and SNP and SxP tests, where they have to choose between two social stimuli with a rather similar value. In contrast, C57BL/6J mice face a motivational conflict at the beginning the SP test, due to their low motivation to investigate the social stimulus, hence experiencing a higher anxiety level at this stage. Despite its simplicity, this conceptual model, which was optimized for only one of the four examined tests, recapitulated the behavioral dynamics of all four tests when the dynamics of social reward were modified from low to high values in C57BL/6J mice and from high to low values in SD rats. Thus, this model confirms the pivotal role played by the dynamics of social motivation in driving the distinct behavioral dynamics of SD rats and C57BL/6J mice in the various tests (Fig. 8o).

## Conclusions

Our study reveals that the two rat and mouse laboratory strains most frequently used in social neuroscience and as genetic backgrounds for animal models of human neuropathological conditions, C57BL/6J mice and SD rats, markedly differ in their social investigation behavior. Specifically, SD male rats show immediate strong motivation to interact with same-sex novel social stimuli, while C57BL/6J mice show only a low level of social motivation at the beginning of an encounter with a novel conspecific. Moreover, these two strains also seem to differ in several aspects of brain activity related to social motivation and behavior. Which of these strains is a better model for human social behavior[62], is an open question, particularly as humans are considered as one of the most social species on earth and since they are highly rewarded by social interactions[63]. Nevertheless, we suggest that researchers studying social behavior in rodents, especially in the context of human disorders, should carefully examine which one of these animal models better fits their specific research questions, and adapt their experimental systems to the social behavior of the selected model.

## Methods

**Animals**. All animals were kept in the animal facility of the University of Haifa under veterinary supervision, with ad libitum access to food (standard chow diet, Envigo RMS, Israel) and water, 23 °C temp, and 60% humidity. Mice subjects were naive C57BL/6J, BALB/c, or ICR (CD-1) adult male or female mice (10–15 weeks), commercially obtained (Envigo, Israel) and housed in groups of 2–5 per cage. Mice stimuli were in-house-grown C57BL/6J, BALB/c, or ICR juvenile male or female mice (21–30 days old), besides the SxP where stimuli were adult female and male C57BL/6J mice (8–12 weeks old). Mice were kept on a 12-h light/12-h dark cycle, lights on at 7 p.m. Rat subjects were SD or Wistar Hannover male or female rats (10–15 weeks), commercially obtained (Envigo, Israel) or grown in-house. Rat stimuli were in-house-grown SD or Wistar Hannover juvenile male or female rats (21–30 days old) commercially obtained (Envigo, Israel), besides the SxP where stimuli were adult female and male SD rats (8–12 weeks old). Rats were kept in groups of 2–5 animals per cage, in a 12-h light/12-h dark cycle, lights on at 9 p.m. Behavioral experiments took place during the dark phase of the animals, under dim red light. All experiments were performed according to the National Institutes of Health guide for the care and use of laboratory animals, and approved by the Institutional Animal Care and Use Committee of the University of Haifa.

**Experimental setups**. Experimental setups for both mice and rats, as well as the video-tracking algorithms and computational analysis software were previously described in detail. Video recording was done using FlyCapture Ver. 2.7.3.18 (FLIR Systems, Wilsonville, OR, USA).

**Behavioral paradigms**. SP/SNP paradigm: The SP/SNP paradigm consisted of a 20-min habituation to the arena, followed by insertion of empty chambers and 15 min of habituation to their presence, during which stimuli were placed in other

chambers for acclimation. Thereafter, social and object (plastic toy, ~5 × 5 cm) stimuli were randomly inserted each to a different chamber, and the SP test was performed for 5 min. Following the SP test, the chambers with the stimuli were removed from the arena, and the subject was left alone for 15 min. Then, the chambers were inserted again, this time to the other two corners of the arena, with one containing the same social stimulus used for the SP test (familiar stimulus) and the other containing a novel stimulus, and the SNP test took place for 5 min. Notably, the familiar stimulus was always placed in a different corner relative to the SP test. At the end of the SNP test, the experimental subject was placed back in its home cage, while the stimuli were left in the chambers for the next experiment or placed back in their home cage at the end of the experimental session.

Sex preference (SxP): The SxP test consisted of 15-min habituation to the arena with empty chambers, followed by exposing the subject for 5 min to both adult female and male stimuli (8–12 weeks old), as done in the SNP test.

Social vs. food paradigm: The social vs. food paradigm consisted of a 15-min habituation to the arena with two empty chambers in it, followed by 5-min exposure to a novel social stimulus (as above), located within a chamber in one corner of the arena, and to a chamber full of food pallets (standard chow diet, Envigo RMS, Israel) in the opposite corner. The food chamber was filled with food pallets to a height of 5 cm and had a metal mesh with 0.5 × 0.5-cm holes, to prevent the animals from consuming the food. The same subject animal (8 rats and 16 mice) performed the test three times: mice—at satiety, 4 h, and 24 h of food deprivation; rats—satiety, 24 h, and 48 h of food deprivation. Between these tests, the animals were returned to their group-housed home cage with free access to water.

**Behavioral analysis**. *Investigation time*: Behavioral analysis was done after correcting the raw behavioral data by considering any gap of <0.5 s in investigation of a given stimulus as part of the same investigation bout. Investigation time was calculated in 20-s bins across all tests.

*Investigation bouts*: In some analyses, we categorized the different investigation bouts according to their length, and calculated investigation time for each duration category. In all these analyses, as well as for drawing behavioral heatmaps, we excluded all bouts made at the last minute of the experiment because of the bias toward observing short bouts during this minute, due to the end of the experiment.

*Relative differential investigation (RDI)*: RDI was defined as the absolute value of the difference in investigation time between the two stimuli, divided by their sum.

*Transitions*: A transition between stimuli was defined as the time point when investigation of a new stimulus (relative to the other stimulus) started. The mean rate of transitions was calculated at 20-s bins.

*Center/periphery ratio*: The center was defined as the inner 30% of the arena.

*Interaction time in free interactions*: To track interactions within dyads of same-age and same-genotype rats, we used custom-made algorithm written in MATLAB (MathWorks, Natick, MA, USA, Ver. 2017a) that tracked the body contours of the animals and determined for each frame whether they are in contact (only one contour is detected = "interaction") or not (two contours are detected = "no interaction"). All video analyses were done after correcting the raw behavioral data (extracted by the above algorithm) by neglecting any gap of 0.5 s in the interaction.

**Measuring stimulus movement by piezoelectric sensors**. *Setup*: Stimulus movements were measured using six piezoelectric ceramic disks (27 mm in diameter) connected in parallel. The disks were evenly distributed along the triangulated Perspex floor and adjusted to it using lamination foil. The signal from the piezodisks was transferred to the analog input of a RHD2000 recording system (Intan Technologies, Los Angeles, CA, USA) through a protective metal tube adjusted to the inner wall of the triangulated chamber (see scheme in Fig. 5b).

*Analysis*: All signals were analyzed using a custom-made MATLAB (2017a) analysis program. The raw signal was recorded at 20 kHz. The signal was then downsampled to 2000 Hz and band-pass filtered between 10 and 100 Hz using a Butterworth filter. Large movements were detected using a threshold of more than 20% of the maximum signal absolute value. Varying this threshold between 20 and 40% did not change the final results. For detecting the subject's tendency for stimulus investigation after a stimulus movement, we analyzed all periods meeting the criteria of no social investigation by the subject and no large movements by the stimulus for at least 3.5 s before a given stimulus movement (Fig. 5e, f, j, k). Varying this period between 2 and 8 s did not change the final results. For statistical analysis (Fig. 5g, l), the total time within 3.5 s after the movement was considered for measuring investigation time.

**c-Fos expression analysis**. Immunohistochemistry was performed on free-floating 50-μm coronal brain sections, which were blocked in 1% Triton-phosphate-buffered saline (PBS) buffer containing 1% normal goat serum for 1 h at room temperature, and then incubated with rabbit monoclonal anti-c-Fos antibody (Cell Signaling, 9F6, Cat. #2250, 1:500) in 1% Triton-PBS buffer for 48 h at 4 °C. The c-Fos signals were revealed using a biotinylated secondary antibody (Biotinylated Goat Anti-Rabbit IgG Antibody (H + L), 1:500) and ABC kit (Vector Laboratories, Burlingame, CA, USA), and subsequently with diaminobenzidine as the chromogen (Sigma-Aldrich). The counting of c-Fos immunopositive cells was

done manually using Fiji software Ver. 20160205 (http://www.imagej.net/Fiji), through the identification of the nuclear brown-reaction products. The quantitative analysis was performed unilaterally and comprised the integral areas of LSV, LSD, and MeA, whereas a representative region of NAc was selected for the examination.

**Computational model**. We used a discrete-time Markov model with dynamic transition probabilities for the simulation (see Table 1). The model has four states (Fig. 8a, b): interacting with stimulus 1 (State 1), interacting with stimulus 2 (State 2), stillness (State 3), and arena exploration (State 4). We used an evolutionary multiobjective optimization algorithm to fit the model parameters using the experimental results of the SP test in mice. Then, we modified the initial values of social reward in the model to fit those expected for the SNP test in mice and SP/SNP tests in rats (see fixed model parameters in Table 2) where the anxiety (a) and the reward for stimuli 1 and 2 ($r_1$, $r_2$) are time dependent:

$$a(t) = a_0 e^{\frac{-t}{\tau_a}} - |r_1 - r_2|, \tag{1}$$

$$r_{1/2}(t) = r_{1/2_0} e^{\frac{sT_{1/2}}{\tau_r}}. \tag{2}$$

$T_{1/2}$—the total time spent exploring stimulus 1 or 2, $s = 1$ for mice and $-1$ for rats.

The MATLAB code of the model can be found in Supplementary Data 2 and GitHub at the following link: [https://github.com/shainetser/Computational-model-of-social-preference-behavior-].

**Statistical analysis**. All averaged data are shown as mean ± SEM values. Statistical tests were performed using SPSS 21.0 or MATLAB (statistical toolbox 2019a). Data normality was tested using the Kolmogorov–Smirnov and Shapiro–Wilk tests. A paired *t* test was used to compare between different conditions or stimuli for the same single group, and an independent *t* test was used to compare a single parameter between two distinct groups. In cases where data were not normally distributed, we used Wilcoxon signed-rank test or Mann–Whitney *U* test. For comparison between multiple groups or parameters, a classical analysis of variance (ANOVA) or Welch's ANOVA tests were applied to the data, depending on homogeneity of variances. For comparison between multiple groups and parameters, a mixed-model ANOVA test was applied. This model contains one random effect (ID), one within effect, one between effect, and the interaction between them. For comparison within a group using multiple parameters, a two-way repeated-measure analysis-of-variance model was applied. This model contains one random effect (ID), two within effects, and the interaction between them. In the case of interaction between the effects in ANOVA, only the interaction values were reported in the text. All ANOVA tests were followed, if the main effect or interaction was found, by post hoc Student's *t* test in the case of classical ANOVA, or with Games–Howell test in the case of Welch's ANOVA. Significance was set at 0.05. The parameters and results of all statistical tests are supplied in Supplementary Data 1.

**Reporting summary**. Further information on research design is available in the Nature Research Reporting Summary linked to this article.

## Data availability
The source data underlying all figures and supplementary figures are provided as a Source Data file. Raw datasets generated and/or analyzed during the current study are available from the corresponding author on reasonable request. Source data are provided with this paper.

## Code availability
All codes used for the current study are available from the corresponding author on reasonable request. The code used for the computational model is publicly available at the following link: [https://github.com/shainetser/Computational-model-of-social-preference-behavior-]. The code used for analyzing the stimulus's movement is available at the following link: [https://github.com/shainetser/PiezoElectricSensors_Code].

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

## Acknowledgements

This study was supported by the Human Frontier Science Program (HFSP grant RGP0019/2015 to V.G. and S.W.), the German Research Foundation (DFG) grants GR 3619/16-1 (to V.G. and S.W.), 3619/7-1, GR 3619/8-1, GR 3619/13-1, GR 3619/15-1, and DFG within the Collaborative research Center SFB 1158-2, Fritz Thyssen foundation Ref. 10.19.1.015MN to V.G., the Israel Science Foundation (ISF grants #1350/12, 1361/17), the Ministry of Science, Technology and Space of Israel (Grant #3-12068), a joint grant of the Ministry of Science, Technology and Space of Israel and the Ministries of Europe and Foreign Affairs (MEAE) and of Higher Education, Research and Innovation (MESRI) of France (Grant #3-16545), and a donation of the Milgrom Family to S.W. The authors thank Thomas Splettstoesser (www.scistyle.com) for his help with the preparation of figures.

## Author contributions

S.N.: conceptualization, methodology, software, validation, formal analysis, data curation, visualization, project administration, and writing—review and editing; A.M.: investigation, visualization, and writing—original draft; H.M.: investigation; A.Z.: conceptualization and software; S.H.Z.: investigation and formal analysis; M.B.: investigation; A.B.: methodology and resources; V.G.: conceptualization, writing—review and editing, supervision, project administration, funding, and acquisition; S.W.: conceptualization, writing—original—draft, supervision, project administration, and funding acquisition.

## Competing interests

The authors declare no competing interests.
