## [Peer Review File · Nature Communications]

Reviewers' comments:

Reviewer #1 (Remarks to the Author):

In this study, Netser et. al. compared the behaviors of rats and mice using a social/object preference task and a novel social/familiar social preference task. They found that mice and rats have different dynamics when exposed to a social target with the rat showing overall higher bias towards a social target than a C57 male mouse. Careful analysis of the stimulus mouse behavior suggests that the stimulus animal do not contribute to the differential responses of the rat vs. mouse to a social target. The authors further advocated the idea that rats are more socially motivated than mice by showing (1) higher Fos induction in the NAc in rats than mice after exposing to a novel social target; (2) Between food and a social stimuli, rats but not mice show a preference towards the social stimulus in satiated state. In general, the paper is interesting and informative, providing a detailed comparison between mice and rats. The simple but smart design of the behavior task (food and social stimulus competition) and careful behavioral analysis is the highlight of the study. Nevertheless, there are several important questions to be addressed before the reviewer is convinced of the conclusion of the paper: rat is more social than mouse.

1. Strain consideration. The SD rat strain is an outbred strain while the C57BL/6 strain is an inbred strain. As such, the individual difference (including the smell) of SD rats is expected to be much larger than that of C57 mice. Comparing the SD rat with C57 mice is kind of comparing the behaviors of identical twins vs. two random strangers. Whether the behavioral difference is due to species or strain difference is an important question to address. Considering that this paper is a pure behavioral study, the authors should address this question instead of leaving it to future studies as mentioned in the discussion. A proper and widely used mouse outbred strain is CD1 (or Swiss Webster). The social behaviors of CD1 (e.g. maternal aggression) in many ways are more similar to wild mice in comparison to C57BL/6.
2. Sex consideration. The study focuses solely on male subjects although the rationale is not clear. As correctly pointed out by the authors, mice are more territorial than rats. Thus, it is perhaps not surprising that male mice and male rats show different level of interest towards another male conspecific. Since territorial behaviors are largely shown by males, it is important to see whether the female rats and mice also differ in their social interest. If not, it will argue against the conclusion that rats are generally more social than mice.
3. Free moving vs. caged animals. The study focuses on behavioral tasks involving a confined social target. While this makes the quantification simpler, it is a less natural condition. Please perform behavioral experiments under free moving conditionings to understand whether the differential social interest remains between mice and rats. Simple parameters such as investigation latency, duration of each bout, interval between investigations could be analyzed.
4. Lack of c-Fos induction in mice after social interaction. It is surprising that the authors observed no increase in c-Fos in various brain regions after social interaction in mice. This is inconsistent with many previous c-Fos studies (e.g. Kim et. al. 2015 Cell Reports). This might be explained by the high level of background c-Fos observed in the control mice, limited access to the stimulus animal or the nature of the stimulus animal. In any case, the authors should investigate and explain this negative result in light

of previous findings.

5. NAc activation. While VTA and NAc are known for its role in reward and motivation, recent studies demonstrate that NAc is heterogeneous and different subregions could be recruited by positive and negative experience (e.g. de Jong JW et. al. 2019 Neuron). Thus, it is important to provide more details regarding the exact regions under investigation.

6. There are some noticeable grammatical and typographical errors. E.g. in the Introduction section, “only few previous studies have directly compared the social behavior...”; in the Results section, “As in the SP test, short bouts did not differ between...” (page 5), “and even following 24 hour starvation,.... Nevertheless, after 48 hour starvation...” (page 17), “we focused on the specific condition were no preference to any of the stimuli ...” (page 18).

Reviewer #2 (Remarks to the Author):

Netser et al. present a novel and important paper, dissecting the crucial differences between mice and rats in social interaction and related neuronal activity responses to social cues. The authors provide important data on the distinction between the two rodent species, concluding that rats display significantly greater motivation for social interaction, and should be considered with higher priority as the model of choice in studies of social behavior. The manuscript is well written, and produces compelling evidence as to the difference between the 2 tested strains. However, some additional data and information are required in order to substantiate the major conclusion of this research:

1. The authors focus on a single mouse strain and a single rat strain. Although these strains are indeed the ones most commonly used, they do not necessarily represent the common behavioral or neurochemical phenotype of their species. In order to faithfully determine the distinction between mice and rats, at least one more commonly used strain of mice (e.g. CD1, BalbC) and one more strain of rats (e.g. Wistar) should be added to the analysis.

2. In addition, if possible I recommend adding additional social behavioral test, conducted in a different setting (e.g. resident intruder,).

3. It is not clear how the classification to short, intermediate and long bouts was performed. Is it based on the literature? Some formula / calculation? This needs to be clarified and justified.

4. In addition, supplementary Fig. 2 suggests that the classification of short bouts to <6 sec was based on the peak time of rats, whereas determination based on the peak of mice (~ 9 sec) might have produced opposite results.

5. The authors state they used different conditions between the groups (i.e. following 15 min of SP for

rats vs. 5 min of SP for mice), since initial testing under the same conditions produced null results for the rats. A reliable comparison between groups requires testing under the same conditions, thus the mice should also be tested following 15 min of SP or explained the reason of testing the two species in different conditions.

6. As for the 5 min of SP condition for the rats (sup Fig. 3A, C), please provide the p value. It appears as if the difference between the novel and familiar is almost significant, but in the opposite direction.

7. The conclusions from figure 4, regarding the lack of effect of the stimuli's movements, should be moderated, as the authors did not perform any direct manipulation.

8. The results of the mice might be attributed to greater sensitivity to stress (under the specific conditions of the current research, e.g. food deprivation, removal from cage). The authors need to add measurements of stress-related behaviors and/or blood cortisol as a control.

9. I suggest that the interpretation of the results would be toned-down, as to avoid excluding mice as a suitable model altogether. Perhaps discussing the advantages and disadvantages of each species, and conclude that the experimental setup and conditions should be carefully adapted to each animal model (as performed by the authors in prolonging the SP duration for the rats).

Minor comments:

10. In my opinion, the figures should be organized such that rats and mice are aligned in the same figure side-by-side, while the different tests (SP, SNP) are separated to different figures.

11. The colors in figure 1F (and equivalents) are barely distinguishable. Please replace the colors or use different markers.

Reviewer #3 (Remarks to the Author):

The authors compared two laboratory rodent species (SD rats and C57Bl/6J mice) for their motivation to seek out a social stimulus under three different conditions. This is a relevant, timely and important comparison because most current translational research relies on the use of common laboratory mouse strains only. The authors demonstrate that there are important differences in social motivation between mice and rats. The main finding is that rats show instant motivation to seek out a social stimulus while mice show a delayed interest in exploring a social stimulus. This knowledge is essential for anyone using either rats or mice because it will affect the interpretation of the animal's social motivation. The authors further show that a novel computational model can simulate this species difference in social motivation. Overall, this manuscript provides novel insights into species differences in the motivation to seek out a social stimulus. However, I have some issues with several elements of the study that are described in

more detail below.

Major issues:

1. Throughout the manuscript, the term “reward” is used as interpretation of the behavioral and neuronal activity findings. But this seems unjustified. The authors don’t provide evidence that investigation of the social stimulus or activation of the NAc reflect reward. Investigation of a social stimulus is certainly a motivated action, but it is unclear whether this is a rewarding action. Furthermore, the authors suggest that the species difference in Fos activation in the MeA and NAc reflects a species difference in reward. However, the NAc is also involved in the modulation of social withdrawal/social avoidance. Therefore, it is unclear whether Fos activation in the NAc equals reward in the current experimental design.
2. The manuscript could benefit from describing the results throughout the manuscript in more specific terms and to avoid unfounded speculation in the discussion. For example, the title is very unspecific. What are “dynamics of social motivation” and it is unclear if this study demonstrates that these dynamics “drive” distinct patterns of social behavior. The abstract lacks specific outcomes. It is unclear how the neural activity patterns are different. The penultimate sentence in the abstract is a bit weak. What are the distinctions in the social motivation systems? The last sentence in the abstract seems highly speculative based on the current study’s outcomes. We don’t know if rats better mimic the rewarding social interactions as we see these in humans. The discussion also includes several statements that seem unjustified. More specific examples are provided below in the minor issues.
3. Both the social preference and social novelty preference tests have stimulus animals behind a partition. Independent of the behavior of the stimulus animals, this partition restricts interactions which may affect the way the experimental animals are motivated to explore the stimuli and this might be different between the two species. In other words, the species difference may reflect one that is brought forward because of the testing conditions. In addition, the main species difference in social motivation is due to a difference in the timing of social exploration that occurs later in the test in mice and much earlier in the test in rats. Thus, mice might be more vigilant than rats in the beginning of exposure to social stimuli. Therefore, identical experimental setups for rats and mice might not be the best approach to study social motivation in each of these species. These issues should be addressed in the discussion.
4. Figs 1 and 2 legends do not correspond with the graphs. Please also explain in more detail the test procedure and what is shown in the graphs. Explain SP and SNP in the legend. The use of both seconds and minutes makes it harder to compare the graphs, just pick one. Graph B and E seem to be in contrast with each other. Graph C is really unclear: What are the dots? The legend suggests that there should be multiple lines, but only 1 line is shown. The Y-axes in graphs D and E are unclear: What is meant by investigation time in seconds if this is supposed to reflect either less than 6 sec bouts or more than 19 second bouts? Shouldn’t the Y-axis represent # or frequency? But then the Y-axis scale doesn’t seem to fit. The black background in Fig 2A is confusing because it could suggest testing in the dark phase, please explain why a dark background was chosen here.

5. Fos in the mouse control group seems relatively high compared to the control group in rats, which may have obscured social-induced Fos in mice. Moreover, the experimental animals were socially housed prior to the 2-min social stimulus exposure and were in the arena for 15 min prior and for 88 min after the social stimulus exposure. These could all be confounds to the social stimulus-induced Fos induction that affected Fos differently in mice versus rats. Finally, the number of rats and mice in the experimental groups for the Fos study is really low. These limitations should be discussed.

6. Fig. 7 requires a better explanation. For example, how is stillness defined? Where is stress coming from? Please describe Fig. 7N in more detail. What is the reason to add anxiety to the model? The opposite of motivation to approach the social stimulus could be indifference rather than anxiety. Mice might prefer to be alone while rats might dislike being alone. These could be variables explaining the species differences as well.

Minor issues:

1. The title and abstract should mention age and sex of the experimental rats and mice.

2. Page 13: "We concluded that the marked differences exhibited between C57BL/6J mice and SD rats in the SP test are largely due to behavioral distinction between the subjects, rather than the stimuli." Please be more specific regarding what is meant by "behavioral distinction."

3. Page 17, last sentence, it seems unlikely that food deprivation of the same length has a similar influence on the motivation to eat in mice versus rats. Mice likely need food sooner than rats.

4. Page 20, "with no possible direct transition between S1, S2, and S4." Why is there no possible transition between S1/S2 and S4, i.e., between stimulus investigation and arena exploration?

5. Discussion, first paragraph, "we observed weaker social recognition memory in rats than in mice, as reflected by the SNP test". The authors tested for social novelty preference and did not really test for social recognition memory and can therefore not make this statement.

6. Page 23, please be more specific in the last sentence of the first paragraph in the discussion.

7. Page 23, when discussing the natural social structure of rats and mice, it will be important to acknowledge that this is referring to the Norway rat and the wild house mouse.

8. The following statements on page 23 require references: "There are substantial differences in the natural social structure of rats and mice. Although both species live in large hierarchical groups, rats are much less territorial and the hierarchy between males is far from absolute." and "As a result, interactions between male mice are less common and are more aggressive and territorial in nature than in rats."

9. References on page 23 appear by name and are missing in the References list.
10. Page 23 “Accordingly, rats are generally considered more social and less aggressive in male-male interactions than mice”. Is this still referring to Norway rats and wild house mice or laboratory strains of rats and mice? A reference would be helpful here.
11. Page 23, last line: “the results of this study suggest that rats show higher tendency to affiliative male-male interactions than mice. To describe the investigation of a social stimulus as “affiliative” seems a stretch.
12. Page 24, “These results suggest that male SD rats are more rewarded by social interactions with other males than C57BL/6J mice, in accordance with our conclusions.” The results could also be interpreted that rats don’t prefer to be alone while mice do. It is unclear whether this represents reward.
13. Page 24, “differences in the dynamics of social motivation.... which creates...” This seems an incorrect cause and consequence relationship that is not tested in this study.
14. Page 24, “These results directly demonstrate that rats show higher motivation for interaction with novel social stimuli as compared to mice.” I’m not sure if these results directly demonstrate this.
15. Page 25, what do the authors mean by “minimized background activity”? And did the referred study determine this increased motivation for social interactions?
16. Page 25, “leaving in place all baseline c-Fos activity”, what do the authors mean by this? How is baseline defined here?
17. Page 25, second paragraph, the MeA and the NAc are also involved in intermale aggression, a motivated behavior that can also be rewarding, but that differs from social play and mother infant bonding in that it is unlikely to be an affiliative behavior.
18. Page 26, “higher reward value of social interactions”. There is not really evidence for reward. It is motivation but it could be induced by a dislike to be alone in rats.
19. Page 27, “Which of them is a better model for human neurodevelopmental disorders associated with impaired social behavior, such as autism spectrum disorder, is an open question.” This seems inappropriate to mention here. What is the link between the current behavioral paradigm and impaired social behaviors as seen in autism?
20. Page 27, “as humans are considered one of the most social species on earth and since social interactions are highly rewarded by them, our study suggests that SD rats are more suitable to model

human social behavior as compared to C57BL/6J mice.” This statement seems unfounded and would require a more thorough and detailed discussion.

We thank the reviewers for their excellent and helpful comments and suggestions. I believe that thanks to them the revised manuscript, in which we did our best to address all the issues raised by the reviewers, is much better than the previous version. Below we detailed the changes we made to the manuscripts and our responses to the reviewers' comments. For reviewers' convenience, the main changes made in the manuscript are highlighted in yellow in the highlighted version of the revised manuscript.

Reviewers' comments:

Reviewer #1 (Remarks to the Author):

In this study, Netser et. al. compared the behaviors of rats and mice using a social/object preference task and a novel social/familiar social preference task. They found that mice and rats have different dynamics when exposed to a social target with the rat showing overall higher bias towards a social target than a C57 male mouse. Careful analysis of the stimulus mouse behavior suggests that the stimulus animal do not contribute to the differential responses of the rat vs. mouse to a social target. The authors further advocated the idea that rats are more socially motivated than mice by showing (1) higher Fos induction in the NAc in rats than mice after exposing to a novel social target; (2) Between food and social stimuli, rats but not mice show a preference towards the social stimulus in satiated state. In general, the paper is interesting and informative, providing a detailed comparison between mice and rats. The simple but smart design of the behavior task (food and social stimulus competition) and careful behavioral analysis is the highlight of the study. Nevertheless, there are several important questions to be addressed before the reviewer is convinced of the conclusion of the paper: rat is more social than mouse.

1. Strain consideration. The SD rat strain is an outbred strain while the C57BL/6 strain is an inbred strain. As such, the individual difference (including the smell) of SD rats is expected to be much larger that of C57 mice. Comparing the SD rat with C57 mice is kind of comparing the behaviors of identical twins vs. two random strangers. Whether the behavioral difference is due to species or strain difference is an important question to address. Considering that this paper is a pure behavioral study, the authors should address this question instead of leaving it to future studies as mentioned in the discussion. A proper and widely used mouse outbred strain is CD1 (or Swiss Webster). The social behaviors of CD1 (e.g. maternal aggression) in many ways are more similar to wild mice in comparison to C57BL/6.

- **Response: We conducted experiments with two more outbred strains: Wistar Hannover rats and ICR mice (CD-1, as recommended by the reviewer). The results, which are included in a new Figure 4 (see below), show that these two strains are in between SD rats and C57BL mice that seem to represent two extremes.**

2. Sex consideration. The study focuses solely on male subjects although the rationale is not clear. As correctly pointed out by the authors, mice are more territorial than rats. Thus, it is perhaps not surprising that male mice and male rats show different level of interest towards another male conspecific. Since territorial behaviors are largely shown by males, it is important to see whether the female rats and mice also differ in their social interest. If not, it will argue against the conclusion that rats are generally more social than mice.

- **Response: We conducted similar experiments in female rat and mice and found similar differences as found between males. These results are displayed in a new supplementary Figure 2 (see below).**

3. Free moving vs. caged animals. The study focuses on behavioral tasks involving a confined social target. While this makes the quantification simpler, it is a less natural condition. Please perform behavioral experiments under free moving conditionings to understand whether the differential social interest remains between mice and rats. Simple parameters such as investigation latency, duration of each bout, interval between investigations could be analyzed.

- **Response: We conducted the requested experiments with freely moving animals and found that even in these conditions rats show significantly higher level of interaction, which is due to higher level of long interactions as found by us using restricted interactions. These results are displayed in a new supplemental Figure 5 (see below) that we added to the manuscript.**

4. Lack of c-Fos induction in mice after social interaction. It is surprising that the authors observed no increase in c-Fos in various brain regions after social interaction in mice. This is inconsistent with many previous c-Fos studies (e.g. Kim et. al. 2015 Cell Reports). This might be explained by the high level of background c-Fos observed in the control mice, limited access to the stimulus animal or the nature of the stimulus animal. In any case, the authors should investigate and explain this negative result in light of previous findings.

- **Response: We think that the difference between our results and previous studies stems from two points. First, the issue of prior interactions, which is considered as background by the reviewer. Unlike other previous studies in which the animals were isolated to significant periods of at least 24 hours to eliminate any background cFos expression and create high social appetitive state in the animals, we took the animals directly from their home cage to perform the SP experiment. This was done because we wanted to examine specifically the cFos expression which is induced by the exposure to a novel stimulus, as opposed to the familiar cagemates, in the same conditions as the behavioral experiments. Thus, our methodology retains the "background" expression, which is elicited by the interactions with the cagemates, and looks only for induction above this relatively high "background" caused by the interaction with the novel social stimulus. The fact that we saw strong induction in rats fits with their behavior and suggest a wave of cFos induction caused by the novelty of the stimulus during the SP test, as opposed to mice. Second, we exposed the animals to an exceptionally short period of interaction (2 min), in which we saw much higher motivation of rats to social interactions as compared to mice.**

However, as the motivation of mice for social interactions seems to increase with time, we predicted that we will see cFos induction in the examined area following this period of interaction, which was used by most previous studies. We therefore repeated the experiment with mice and added a group with 5-min exposure. As predicted by us, while the 2-min exposure was again insufficient to induce cFos expression, 5-min exposure did cause significant induction of cFos expression, thus confirming our prediction. These data are shown in a new supplementary Figure 7 (see below) and widely discussed in the Discussion section of the revised manuscript.

5. NAc activation. While VTA and NAc are known for its role in reward and motivation, recent studies demonstrate that NAc is heterogeneous and different subregions could be recruited by positive and negative experience (e.g. de Jong JW et. al. 2019 Neuron). Thus, it is important to provide more details regarding the exact regions under investigation.

- **Response: We analyzed the previous data separately for the core and shell of the NAc and found no difference from the results analyzed for the entire NAc. Yet, our new data which appears in supplemental Figure 7 of the revised manuscript (shown above), where we separately analyzed the NAc core and shell, show that after 5 min exposure to a novel social stimulus there is a significant c-Fos induction in the shell, but not the core, of the NAc.**

6. There are some noticeable grammatical and typographical errors. E.g. in the Introduction section, “ only few previous studies have directly compared the social behavior...”; in the Results section, “ As in the SP test, short bouts did not differ between...” (page 5), “ and even following 24 hour starvation,... Nevertheless, after 48 hour starvation...” (page 17), “ we focused on the specific condition were no preference to any of the stimuli ...” (page 18).

- **Response: These errors and others were corrected in the revised manuscript.**

Reviewer #2 (Remarks to the Author):

Netser et al. present a novel and important paper, dissecting the crucial differences between mice and rats in social interaction and related neuronal activity responses to social cues. The authors provide important data on the distinction between the two rodent species, concluding that rats display significantly greater motivation for social interaction, and should be considered with higher priority as the model of choice in studies of social behavior. The manuscript is well written, and produces compelling evidence as to the difference between the 2 tested strains. However, some additional data and information are required in order to substantiate the major conclusion of this research:

1. The authors focus on a single mouse strain and a single rat strain. Although these strains are indeed the ones most commonly used, they do not necessarily represent the common behavioral or neurochemical phenotype of their species. In order to faithfully determine the distinction between mice and rats, at least one more commonly used strain of mice (e.g. CD1, BalbC) and one more strain of rats (e.g. Wistar) should be added to the analysis.

- **Response: As stated above (in response to point 1 of reviewer#1), we conducted experiments with two more outbred strains: Wistar Hannover rats and ICR (CD-1, as recommended by the reviewer) mice. The results, which are included in a new figure (Fig. 4), show that these two strains are in between SD rats and C57BL mice that seem to represent two extremes.**

2. In addition, if possible I recommend adding additional social behavioral test, conducted in a different setting (e.g. resident intruder,).

- **Response: We added another social behavior test, the sex preference test, and got similar results to the SNP test. These results, which are presented as a supplementary Figure 4 (see below) in the revised manuscript, show that the SP test is an exception, a point that we further discuss.**

3. It is not clear how the classification to short, intermediate and long bouts was performed. Is it based on the literature? Some formula / calculation? This needs to be clarified and justified.

- **Response:** We added a detailed explanation for this type of categorization in the legend of supplementary Figure 1 (see below) that presents the distributions of bouts according to their length.

Supplementary Figure 1. Distributions of investigation bout durations during the SP and SNP tests

- A) Superimposed distributions of bout duration for C57BL/6J mice (n=58) during the SP test for the social (blue) or object (light blue) stimulus.
- B) As in A, for SD rats (n=60). Dashed lines represent the borders between the populations of short (<6 sec), intermediate (>6 sec, < 19 sec) and long (>19 sec) investigation bouts. Both borders mark clear deeps in the distributions of social investigations in both mice and rats, thus separate well-define populations.
- C) As in A, for the SNP test with the novel (red) and familiar (Blue) social stimuli. Dashed lines represent the same values as in A, although the distinct populations of investigation bouts are not well-defined in the SNP test.

As in C, for rats.

4. In addition, supplementary Fig. 2 suggests that the classification of short bouts to <6 sec was based on the peak time of rats, whereas determination based on the peak of mice (~ 9 sec) might have produced opposite results.

- **Response: As detailed in the figure legend above, in the distributions of bout durations of both rats and mice SP tests there is a clear deep at 6 sec which defines a clear population of short bouts below 6 sec. A similar deep**

is observed at 19 sec, seen better in mice, which defines the population of long bouts.

5. The authors state they used different conditions between the groups (i.e. following 15 min of SP for rats vs. 5 min of SP for mice), since initial testing under the same conditions produced null results for the rats. A reliable comparison between groups requires testing under the same conditions, thus the mice should also be tested following 15 min of SP or explained the reason of testing the two species in different conditions.

- **Response: We tested the animals in similar conditions, which are the minimal exposure time during the SP test that yielded clear preference in the SNP test. Moreover, the argument of the reviewer would be justified, in our eyes, if indeed we were reporting differences between rats and mice in the SNP test following different exposure times. However, there is no difference between the strains in this test and the manuscript is therefore focused on the differences found during the SP test, where there is no difference in the conditions between the animals. Finally, we checked mice with 15 min SP test and there was no change in the results.**

6. As for the 5 min of SP condition for the rats (sup Fig. 3A, C), please provide the p value. It appears as if the difference between the novel and familiar is almost significant, but in the opposite direction.

- **Response: that is right. We provide the p value in the revised manuscript.**

7. The conclusions from figure 4, regarding the lack of effect of the stimuli's movements, should be moderated, as the authors did not perform any direct manipulation.

- **Response: We rephrased the conclusion according to the reviewer note, which is now sounds as "Thus, both rats and mice seem to adjust their investigation behavior to the movements of the social stimulus. However, while rat subjects seem to be attracted by movements of the social stimulus, mouse subjects seem to be deterred by them."**

8. The results of the mice might be attributed to greater sensitivity to stress (under the specific conditions of the current research, e.g. food deprivation, removal from cage). The authors need to add measurements of stress-related behaviors and/or blood cortisol as a control.

- **Response: We added a center/periphery analysis of the movement of subjects during the beginning of habituation in the empty arena, which serves her as an open field test. We found the mice actually showed a slightly lower level of stress than rats. These results are brought in the revised manuscript towards the end of page 5, as quoted below: "...we used the 20-min habituation period preceding the SP test as an open-field test. We calculate the center/periphery location ratio as a measure for**

stress: the higher the ratio, the lower the stress it reflects. We found that C57BL/6J mice exhibited higher center/periphery ratio than SD rats (Mean±SD: mice - 0.22±0.22, rats - 0.08±0.05; t-test, t=3.251, df=71, p<0.001), thus excluding the possibility that a higher level of stress makes them less social than SD rats."

9. I suggest that the interpretation of the results would be toned-down, as to avoid excluding mice as a suitable model altogether. Perhaps discussing the advantages and disadvantages of each species, and conclude that the experimental setup and conditions should be carefully adapted to each animal model (as performed by the authors in prolonging the SP duration for the rats).

- **Response: We have done as requested by the reviewer. Now the last sentence of the conclusion paragraph of the Discussion section (page 20 of the revised manuscript) goes like that:"Nevertheless, we suggest that researchers studying social behavior in rodents, especially in the context of human disorders, should carefully examine which one of these models better fits their specific research questions and adapt their experimental systems to the social behavior of the selected model."**

Minor comments:

10. In my opinion, the figures should be organized such that rats and mice are aligned in the same figure side-by-side, while the different tests (SP, SNP) are separated to different figures.

- **Response: This was an excellent advice, and indeed we reorganized the figures (Figs 1-4) of the revised manuscript as suggested, with rats and mice data presented side by side.**

11. The colors in figure 1F (and equivalents) are barely distinguishable. Please replace the colors or use different markers.

- **Response: we apologize but we don't understand what is the problem with the colors.**

Reviewer #3 (Remarks to the Author):

The authors compared two laboratory rodent species (SD rats and C57Bl/6J mice) for their motivation to seek out a social stimulus under three different conditions. This is a relevant, timely and important comparison because most current translational research relies on the use of common laboratory mouse strains only. The authors demonstrate that there are important differences in social motivation between mice and rats. The main finding is that rats show instant motivation to seek out a social stimulus while mice show a delayed interest in exploring a social stimulus. This knowledge is essential for anyone using either rats or mice because it will affect the interpretation of the animal's social motivation. The authors further show that a novel computational model can simulate this species difference in social motivation. Overall, this manuscript provides novel insights into species differences in the motivation to seek out a social stimulus. However, I have some issues with several elements of the study that are described in more detail below.

Major issues:

1. Throughout the manuscript, the term “reward” is used as interpretation of the behavioral and neuronal activity findings. But this seems unjustified. The authors don't provide evidence that investigation of the social stimulus or activation of the NAc reflect reward. Investigation of a social stimulus is certainly a motivated action, but it is unclear whether this is a rewarding action. Furthermore, the authors suggest that the species difference in Fos activation in the MeA and NAc reflects a species difference in reward. However, the NAc is also involved in the modulation of social withdrawal/social avoidance. Therefore, it is unclear whether Fos activation in the NAc equals reward in the current experimental design.

•

Response: We think that today, following multiple recent studies some of which are cited in the discussion of our manuscript, it is well accepted that social investigation is a rewarded action in rodents. However, to fulfill the reviewer's request, we downplayed our use of the term reward throughout the revised manuscript, and we mainly use the term social motivation.

2. The manuscript could benefit from describing the results throughout the manuscript in more specific terms and to avoid unfounded speculation in the discussion. For example, the title is very unspecific. What are “dynamics of social motivation” and it is unclear if this study demonstrates that these dynamics “drive” distinct patterns of social behavior. The abstract lacks specific outcomes. It is unclear how the neural activity patterns are different. The penultimate sentence in the abstract is a bit weak. What are the distinctions in the social motivation systems? The last sentence in the abstract seems highly speculative based on the current study's outcomes. We don't know if rats better mimic the rewarding social interactions as we see these in humans. The discussion also includes several statements that seem unjustified. More specific examples are provided below in the minor issues.

- **Response: We did our best to fulfill this request in the revised manuscript.**
- **However, in some cases we disagree with the reviewer regarding what is considered unspecific term. For example, "dynamics of social motivation" is**

specifically referring to changes across time in the motivation for social interaction.

- **In the revised manuscript we show evidence, using the social-food competition (see Figure 7F), that the changes in motivation drive changes in the dynamics of social behavior.**
- **We modified the abstract to make it more specific. Yet, we are highly limited by the word number restrictions.**
- **We omitted the last sentence from the abstract of the revised manuscript.**
- **We omitted the phrase "distinctions in the social motivation systems" from the abstract.**

3. Both the social preference and social novelty preference tests have stimulus animals behind a partition. Independent of the behavior of the stimulus animals, this partition restricts interactions which may affect the way the experimental animals are motivated to explore the stimuli and this might be different between the two species. In other words, the species difference may reflect one that is brought forward because of the testing conditions.

- **Response: This may be claimed against every type of experimental condition, which is always artificial to some extent.**
- **Still, to address the reviewer's concern, we conducted experiments using free interactions between subjects and stimuli and showed that even in this condition, rats still show higher motivation for social interactions. These results appear as a supplementary Figure 5 in the revised manuscript (see response to point #3 of reviewer #1).**

In addition, the main species difference in social motivation is due to a difference in the timing of social exploration that occurs later in the test in mice and much earlier in the test in rats. Thus, mice might be more vigilant than rats in the beginning of exposure to social stimuli. Therefore, identical experimental setups for rats and mice might not be the best approach to study social motivation in each of these species. These issues should be addressed in the discussion.

- **Response: We addressed this issue at the concluding paragraph of the Discussion section (page 20), stating that: "Nevertheless, we suggest that researchers studying social behavior in rodents, especially in the context of human disorders, should carefully examine which one of these models better fits their specific research questions and adapt their experimental systems to the social behavior of the selected model."**

4. Figs 1 and 2 legends do not correspond with the graphs. Please also explain in more detail the test procedure and what is shown in the graphs.

- **Response: Done.**

Explain SP and SNP in the legend.

- **Response: Done.**

The use of both seconds and minutes makes it harder to compare the graphs, just pick one.

- **Response: Done.**

Graph B and E seem to be in contrast with each other.

- **Response: These graphs are not contrasting, first of all because graph B shows all the investigation time, regardless of bout duration, while graph E shows the investigation time only the extended bouts (>19 sec). It is true that as reflected in B, the animals are doing overall more social investigation at the beginning than at the end, but this is mainly due to short and intermediate (<19 sec) bouts, while extended bouts are low at the beginning and high towards the end.**

Graph C is really unclear: What are the dots? The legend suggests that there should be multiple lines, but only 1 line is shown.

- **Response: We clarified the explanation of the graph, stating: "Each punctum denotes the beginning of investigation of a new stimulus, and each row represent a single subject. The mean rate (using 20-sec bins) is denoted by the red line (right red Y-axis)."**

The Y-axes in graphs D and E are unclear: What is meant by investigation time in seconds if this is supposed to reflect either less than 6 sec bouts or more than 19 second bouts?

- **Response: We changed the title of the axis to "Pooled time" make it clearer that this is the time pooled from all bouts which are <6 sec or >19 sec.**

Shouldn't the Y-axis represent # or frequency? But then the Y-axis scale doesn't seem to fit.

- **Response: No, in this graphs we present the investigation time accumulating during the various types of bouts. Principally we could also present the number of events, which is qualitatively similar to the accumulating time. However, since the number of extended bouts (>19 sec) is relatively low this type of presentation is very noisy.**

The black background in Fig 2A is confusing because it could suggest testing in the dark phase, please explain why a dark background was chosen here.

- **Response: We used dark phase for the rats just to make it similar to the reality, where we tested while rats on dark background. To avoid confusion, we eliminated these cartoons, which are not very informative.**

5. Fos in the mouse control group seems relatively high compared to the control group in rats, which may have obscured social-induced Fos in mice. Moreover, the experimental animals were socially housed prior to the 2-min social stimulus exposure and were in the arena for 15 min prior and for 88 min after the social stimulus exposure. These could all be confounds to the social stimulus-induced Fos induction that affected Fos differently in mice versus rats. Finally, the number of rats and mice in the experimental groups for the Fos study is really low. These limitations

should be discussed.

- **Response: We explained this issues above in response to point #4 of reviewer #1. As stated there, unlike other previous studies in which the animals were isolated to significant periods of at least 24 hours to eliminate any background cFos expression and create high social appetitive state in the animals, we took the animals directly from their home cage to perform the SP experiment. This was done because we wanted to examine specifically the cFos expression which is induced by the exposure to a novel stimulus, as opposed to the familiar cagemates, in the same conditions as the behavioral experiments. Thus, our methodology retains the "background" expression, which is elicited by the interactions with the cagemates, and looks only for induction above this relatively high "background" caused by the interaction with the novel social stimulus. The fact that we saw strong induction in rats fits with their behavior and suggest a wave of cFos induction caused by the novelty of the stimulus during the SP test, as opposed to mice. Second, we exposed the animals to an exceptionally short period of interaction (2 min), in which we saw much higher motivation of rats to social interactions as compared to mice. However, as the motivation of mice for social interactions seems to increase with time, we predicted that we will see cFos induction in the examined area following this period of interaction, which was used by most previous studies. We therefore repeated the experiment with mice and added a group with 5-min exposure. As predicted by us, while the 2-min exposure was again insufficient to induce cFos expression, 5-min exposure did cause significant induction of cFos expression, thus confirming our prediction. These data are shown in a new supplementary Figure 7 (see below) and widely discussed in the Discussion section of the revised manuscript.**
- **As for the number of animals, we repeated the experiment with mice and got exactly the same results. All this issues are addressed in the discussion (2nd paragraph of page 17), as requested.**

6. Fig. 7 requires a better explanation. For example, how is stillness defined? Where is stress coming from? Please describe Fig. 7N in more detail. What is the reason to add anxiety to the model? The opposite of motivation to approach the social stimulus could be indifference rather than anxiety. Mice might prefer to be alone while rats might dislike being alone. These could be variables explaining the species differences as well.

- **Response: We added explanations for all model parameters in the Results and Discussion sections. We also described the cartoon with more details in the figure legend. As for the parameters, we tried to minimize the number of parameters, otherwise we may get into overfitting of the problem with the model. Therefore, we used only the parameter of anxiety as negatively affecting the animal's decision to investigate the stimuli. Other factors may certainly apply, hence we explicitly state that this model is simplistic. Nevertheless, the preference to be alone or not is actually hidden in the**

reward of the social stimulus.

Minor issues:

1. The title and abstract should mention age and sex of the experimental rats and mice.

- **Response: Done**

2. Page 13: “We concluded that the marked differences exhibited between C57BL/6J mice and SD rats in the SP test are largely due to behavioral distinction between the subjects, rather than the stimuli.” Please be more specific regarding what is meant by “behavioral distinction.”

- **Response: We rephrased the sentence to "We concluded that the marked differences between these two strains in the SP test are mainly due to distinctions between the subjects, rather than the stimuli".**

3. Page 17, last sentence, it seems unlikely that food deprivation of the same length has a similar influence on the motivation to eat in mice versus rats. Mice likely need food sooner than rats.

- **Response: We dealt with this issue in the discussions and referred to the only study we found that directly compared the metabolic state of rats and mice during starvation:**

Menahan, L. A. & Sobocinski, K. A. Comparison of carbohydrate and lipid metabolism in mice and rats during fasting. *Comp Biochem Physiol B* 74, 859-864, doi:10.1016/0305-0491(83)90157-8 (1983).

This study found that both the reduction in body weight (in percentage) and the decrease in plasma insulin level, following 24 hours of fasting were rather similar between rats and mice. Therefore, although we agree that mice likely need food sooner than rats, this could not explain huge difference in their preference of social vs. food during starvation.

4. Page 20, “with no possible direct transition between S1, S2, and S4.” Why is there no possible transition between S1/S2 and S4, i.e., between stimulus investigation and arena exploration?

- **Response: We added an explanation to the revised manuscript, stating that "For simplification, we assumed that every motivated behavior (S1, S2 and S4) has to end at stillness (S3), before a new motivated behavior may arise. Thus, there was no possible direct transition between S1, S2, and S4." As stated, this was done for simplification of the model, otherwise we would have a very large number of probabilities of transitions between states.**

5. Discussion, first paragraph, “we observed weaker social recognition memory in rats than in mice, as reflected by the SNP test”. The authors tested for social novelty preference and did not really test for social recognition memory and can therefore not make this statement.

- **Response: We rephrased this sentence, stating that : " First, we observed weaker social novelty preference in SD rats than in C57BL/6J mice (Fig. 2) that may be related to differences between rats and mice in social recognition memory,"**

6. Page 23, please be more specific in the last sentence of the first paragraph in the discussion.

- **Response: Done.**

7. Page 23, when discussing the natural social structure of rats and mice, it will be important to acknowledge that this is referring to the Norway rat and the wild house mouse.

- **Response: Done.**

8. The following statements on page 23 require references: “There are substantial differences in the natural social structure of rats and mice. Although both species live in large hierarchical groups, rats are much less territorial and the hierarchy between males is far from absolute.” and “As a result, interactions between male mice are less common and are more aggressive and territorial in nature than in rats.”

- **Response: Done.**

9. References on page 23 appear by name and are missing in the References list.

- **Response: Corrected.**

10. Page 23 “Accordingly, rats are generally considered more social and less aggressive in male-male interactions than mice”. Is this still referring to Norway rats and wild house mice or laboratory strains of rats and mice? A reference would be helpful here.

- **Response: We clarified this sentence and brought a reference (page 15, 1st paragraph).**

11. Page 23, last line: “the results of this study suggest that rats show higher tendency to affiliative male-male interactions than mice. To describe the investigation of a social stimulus as “affiliative” seems a stretch.

- **Response: The referred study measured the distance between the animals, not investigations. Thus, affiliative seems appropriate to describe this type of behavior.**

12. Page 24, “These results suggest that male SD rats are more rewarded by social interactions with other males than C57BL/6J mice, in accordance with our conclusions.” The results could also be interpreted that rats don’t prefer to be alone while mice do. It is unclear whether this represents reward.

- **Response: We changed the term "rewarded" to "attracted".**

13. Page 24, “differences in the dynamics of social motivation... which creates...” This seems an incorrect cause and consequence relationship that is not tested in this study.

- **Response: This sentence is just a suggestion explain the results, not a conclusion. We rephrased it to make this point clear as “These differences may be due to distinctions between the two strains in the dynamics of their motivation for social interaction that creates.”.**

14. Page 24, “These results directly demonstrate that rats show higher motivation for interaction with novel social stimuli as compared to mice.” I’m not sure if these results directly demonstrate this.

- **Response: We rephrased and omitted to word "directly".**

15. Page 25, what do the authors mean by “minimized background activity”? And did the referred study determine this increased motivation for social interactions?

- **Response: We rephrased the whole paragraph to make the issue of baseline activity clearer and added a reference showing that social isolation increases the motivation for social interaction. It is now written like that: "It should be noted, however, that in all these studies the subjects were kept in social isolation for at least one week before the social encounter, a condition that increases their motivation for social interactions ³⁹."**

16. Page 25, “leaving in place all baseline c-Fos activity”, what do the authors mean by this? How is baseline defined here?

- **Response: We explain in the revised manuscript (page 16, 2nd paragraph) that the term baseline refer to the control condition were no interaction with a social stimulus occurred.**

17. Page 25, second paragraph, the MeA and the NAc are also involved in intermale aggression, a motivated behavior that can also be rewarding, but that differs from social play and mother infant bonding in that it is unlikely to be an affiliative behavior.

- **Response: We agree, however we don't talk about affiliative behavior in this paragraph, just on motivated social behavior.**

18. Page 26, “higher reward value of social interactions”. There is not really evidence for reward. It is motivation but it could be induced by a dislike to be alone in rats.

- **Response: We rephrased this sentence to deal with motivation rather than with reward**

19. Page 27, “Which of them is a better model for human neurodevelopmental disorders associated with impaired social behavior, such as autism spectrum disorder,

is an open question.” This seems inappropriate to mention here. What is the link between the current behavioral paradigm and impaired social behaviors as seen in autism?

- **Response: In this study, we used tests (SP/SNP) which are used in almost all studies of animal models of autism, especially those that present new models. Therefore, we believe that our results are highly relevant for such studies. Nevertheless, to avoid confusion, we omitted direct reference to autism in this paragraph.**

20. Page 27, “as humans are considered one of the most social species on earth and since social interactions are highly rewarded by them, our study suggests that SD rats are more suitable to model human social behavior as compared to C57BL/6J mice.” This statement seems unfounded and would require a more thorough and detailed discussion.

- **Response: We rephrased this sentence to mute down the criticized claim. Now it sounds as " Which of these strains is a better model for human social behavior ⁵⁶, is an open question, especially as humans are considered as one of the most social species on earth and since they are highly rewarded by social interactions ⁵⁷. Nevertheless, we suggest that researchers studying social behavior in rodents, especially in the context of human disorders, should carefully examine which one of these models better fits their specific research questions and adapt their experimental systems to the social behavior of the selected model."**

REVIEWER COMMENTS

Reviewer #1 (Remarks to the Author):

I appreciate that the authors conducted all the experiments suggested by the reviewer. However, in light of the new data showing that Wistar Hannover rats and CD1 mice show very similar social/object preference (none of the analyzed parameters revealed a difference between these two groups), I do not think that it is proper to conclude in the rest of the paper that there is a difference in social behaviors between mice and rats. The paper reveals a deficit in social behaviors of C57BL/6 mice but not a difference between mice and rats in general. I highly suggest to show data for Wistar Hannover rats and CD1 mice for all figures. The comparison between CD1 and C57 is in fact more informative than C57 mice and SD rats as the later involves two variables (strain and species). The authors should use 2-way ANOVA (strain x species) in Figures 4I-4K to understand whether there is in fact any species difference. While the reviewer continues to think the paper is interesting, informative and worth publishing, it is important that the data is presented and interpreted properly to not mislead the field.

Reviewer #2 (Remarks to the Author):

The authors answered all my comments.

Reviewer #3 (Remarks to the Author):

The authors made substantial improvements on the manuscript by incorporating many suggestions from the reviewers and by adding the requested experiments.

However, there are several remaining issues:

1. I would argue that the term social behavior in the title is too broad and should be narrowed to what is studied in the manuscript, which is social investigation behavior or social approach behavior. Likewise, the abstract indicates that social behavior is compared between rats and mice, but it would be more appropriate to indicate what specific form of social behavior is compared, namely social investigation or social approach.

2. The authors indicate in their rebuttal that “However, in some cases we disagree with the reviewer regarding what is considered unspecific term. For example, 'dynamics of social motivation' is specifically referring to changes across time in the motivation for social interaction.”

But this definition is nowhere found in the manuscript. The word dynamics can mean so many different things. So, please include this definition upon first use in the manuscript.

3. There is no consistency in the order of rats and mice throughout the manuscript. Because the title mentions rats first and then mice, please keep this order throughout the manuscript including figures (or

vice versa, but at least be consistent).

4. In figs, please always mention strain of rats and mice (for example, Fig 3 indicates rats and mice)

5. I suggest merging the Wistar and CD-1 data into Fig 1 and providing the outcomes as rationale to pursue the rest of the study with the two extremes, namely SD and C57.

6. The addition of Wistar and CD-1 data is very important and provides a better comparison in social motivation between rats and mice. Therefore, briefly discuss the implications of the Wistar and CD-1 data in the discussion.

7. It is unclear what evidence the following statement is based on as it seems more of a personal statement. Please add references to provide support for this statement: "We presumed that a need to choose between two similar stimuli creates a dilemma and increase the anxiety level of the subject, whereas a strong difference between the stimuli will solve the dilemma and reduce the subject's anxiety" (Page 20 and also similar assumption on page 14)

8. Likewise, this statement seems more a personal opinion rather than a justified definition of anxiety: "the anxiety is a more complex parameter that reflects both the stress level and difference in reward between the two stimuli" (page 20). Please add references to justify this statement.

9. Page 20: "C57BL/6J mice and SD rats, markedly differ in their social behavior"

This is too broad. Please narrow down to social approach or social investigation behavior or social motivation. Any other social behaviors are not determined in the current study and thus, no conclusions can be drawn regarding species differences in general social behavior.

10. Abstract: The authors indicate that "mice and rats are used to model human social behavior". I think I know what the authors mean, but this wording less likely represents that. This reads as if rodents display forms of human social behavior. The authors probably want to convey that rodents are used as model organisms to better understand social behavior and its neural underpinnings, which in turn can be informative to understand forms of human social behavior.

11. Abstract: I think that "Equivalently valid as animal models" is the incorrect phrasing. The question here is whether it matters if you use a specific rat or mouse strain as model organism.

12. I propose to add the following to the last sentence of the abstract: which should be taken into consideration when selecting rats or mice as model organism.

13. Results, page 5: Using the center/periphery location may be a very weak, if at all, measure of stress, unless the authors can find literature suggesting so. In my experience, this is rather a measure of risk

assessment and perhaps anxiety-related behavior, but that does not equal stress. It is also unclear how this non-social measure relates to social preference and interactions.

Pont-by-point response to the reviewers' comments

We thank the reviewers for their excellent and helpful comments and suggestions. We believe that thanks to them the revised manuscript, in which we did our best to address all the issues raised by the reviewers, is much better than the previous version. Below we detailed the changes we made to the manuscripts and our responses to the reviewers' comments. For reviewers' convenience, the main changes made in the manuscript are in Track Changes mode.

Reviewer #1 (Remarks to the Author):

I appreciate that the authors conducted all the experiments suggested by the reviewer. However, in light of the new data showing that Wistar Hannover rats and CD1 mice show very similar social/object preference (none of the analyzed parameters revealed a difference between these two groups), I do not think that it is proper to conclude in the rest of the paper that there is a difference in social behaviors between mice and rats. The paper reveals a deficit in social behaviors of C57BL/6 mice but not a difference between mice and rats in general.

Answer: We thank the reviewer for this very correct point. Indeed, we now conclude that the differences are strain-specific rather than species-specific and this is explicitly stated in the abstract, referred to in the results section (top of page 6) and discussed in the Discussion section (top of page 17).

I highly suggest to show data for Wistar Hannover rats and CD1 mice for all figures. The comparison between CD1 and C57 is in fact more informative than C57 mice and SD rats as the later involves two variables (strain and species).

Answer: In this issue we adopted the suggestion by reviewer #3, to show the data of the SP test for all four strains at the beginning of the Results section and then to focus on the two extremes, C57BL/6J mice and SD rats (last sentence of 1st paragraph of page 7) for the rest of the manuscript. We therefore decided not to show data of more tests for Wistar rats and CD-1 mice, as, to our mind, they may confuse the readers and obscure the focus of the paper, which is on the SP test.

The authors should use 2-way ANOVA (strain x species) in Figures 4I-4K to understand whether there is in fact any species difference.

Answer: While the reviewer is absolutely correct in principle, in our case we had to use Kruskal Wallis test instead of ANOVA because our data in this section violate the “normal distribution of the dependent variable” assumption, and therefore we had to use a non-parametric test which is equivalent to the parametric ANOVA. Since there is no non-parametric equivalent for 2-way ANOVA, we tested our 2 strains and 2 species as 4 independent groups in order to conduct the non-parametric Kruskal Wallis test (one-way ANOVA on ranks). Thereafter, we performed as post hoc the Mann-Whitney u test to compare between the strains and species. Thus, we cannot judge based on the current data if there is a species-dependent difference. We hope that in future studies, using other strains and tests, we would be able to conclude regarding this point.

While the reviewer continues to think the paper is interesting, informative and worth publishing, it is important that the data is presented and interpreted properly to not mislead the field.

Answer: We fully agree with the respected reviewer and hope that the changes we made to the manuscript according to the reviewers' helpful notes will facilitate proper interpretation of our results.

Reviewer #2 (Remarks to the Author):

The authors answered all my comments.

We thank the respectful reviewer.

Reviewer #3 (Remarks to the Author):

The authors made substantial improvements on the manuscript by incorporating many suggestions from the reviewers and by adding the requested experiments.

However, there are several remaining issues:

1. I would argue that the term social behavior in the title is too broad and should be narrowed to what is studied in the manuscript, which is social investigation behavior or social approach behavior. Likewise, the abstract indicates that social behavior is compared

between rats and mice, but it would be more appropriate to indicate what specific form of social behavior is compared, namely social investigation or social approach.

Answer: This is correct and we modified the title as the reviewer suggested, choosing “social investigation behavior” from the two variants proposed by her/him. The abstract was also rephrased as proposed by the reviewer as well as all other relevant sentences in the text.

2. The authors indicate in their rebuttal that “However, in some cases we disagree with the reviewer regarding what is considered unspecific term. For example, 'dynamics of social motivation' is specifically referring to changes across time in the motivation for social interaction.”

But this definition is nowhere found in the manuscript. The word dynamics can mean so many different things. So, please include this definition upon first use in the manuscript.

Answer: The respected reviewer is absolutely right, hence we defined "dynamics" in the Background section of the revised manuscript (last paragraph of page 3), writing "Here we employed this system to compare the temporal pattern (henceforth termed dynamics) of social behavior between the two rodent strains..."

3. There is no consistency in the order of rats and mice throughout the manuscript. Because the title mentions rats first and then mice, please keep this order throughout the manuscript including figures (or vice versa, but at least be consistent).

Answer: We thank the reviewer for this important recommendation. Since the first figures show mice data before rat data we decided to keep this order throughout the manuscript. Therefore, we replace the positions of C57BL/6J mice and SD rats in the title, changed all figures (including supplementary figures) such that mice data are shown first, and adapted the text accordingly.

4. In figs, please always mention strain of rats and mice (for example, Fig 3 indicates rats and mice)

Answer: We corrected all figures to mentions the animal strains, as suggested by the reviewer.

5. I suggest merging the Wistar and CD-1 data into Fig 1 and providing the outcomes as rationale to pursue the rest of the study with the two extremes, namely SD and C57.

Answer: This was an excellent suggestion and we corrected the manuscript accordingly. Yet, we did not merge the CD-1 and Wistar data with Fig. 1, but rather changed the order of figures, such that these data now appears in Fig. 3, before dealing with the SNP (Fig. 4) and other tests. We then explicitly stated that "Thus, it seems as if the differences observed by us in the behavioral dynamics during the SP test are strain-specific, with C57BL/6J mice and SD rats representing two extremes. We therefore focused the rest of the study on these two strains" (end of 1st paragraph of page 7).

6. The addition of Wistar and CD-1 data is very important and provides a better comparison in social motivation between rats and mice. Therefore, briefly discuss the implications of the Wistar and CD-1 data in the discussion.

Answer: As rightly requested by the reviewer, this issue is now discussed within the Discussion section, in the last paragraph of page 18

7. It is unclear what evidence the following statement is based on as it seems more of a personal statement. Please add references to provide support for this statement: "We presumed that a need to choose between two similar stimuli creates a dilemma and increase the anxiety level of the subject, whereas a strong difference between the stimuli will solve the dilemma and reduce the subject's anxiety" (Page 20 and also similar assumption on page 14)

8. Likewise, this statement seems more a personal opinion rather than a justified definition of anxiety: "the anxiety is a more complex parameter that reflects both the stress level and difference in reward between the two stimuli" (page 20). Please add references to justify this statement.

Answer: (to both points 7 and 8): The reviewer is absolutely right about that. To address his questions, we rephrased this statement, which now appears in page 20 as: "Anxiety was defined as a more complex parameter⁶⁰ that in our case reflects both the stress level and the difference in reward between the two stimuli (Fig, 8B). We presumed that the need to choose between two stimuli with similar value will create a motivational conflict and increase the anxiety level of the subject, in accordance with multiple studies associating motivational conflict and uncertainty with anxiety^{27,28,61}."

This statement is supported by the following references"

- 27 Barker, T. V., Buzzell, G. A. & Fox, N. A. Approach, avoidance, and the detection of conflict in the development of behavioral inhibition. *New Ideas Psychol* **53**, 2-12, doi:10.1016/j.newideapsych.2018.07.001 (2019).
- 28 Schlund, M. W. *et al.* The tipping point: Value differences and parallel dorsal-ventral frontal circuits gating human approach-avoidance behavior. *Neuroimage* **136**, 94-105, doi:10.1016/j.neuroimage.2016.04.070 (2016).
- 60 Grupe, D. W. & Nitschke, J. B. Uncertainty and anticipation in anxiety: an integrated neurobiological and psychological perspective. *Nat Rev Neurosci* **14**, 488-501, doi:10.1038/nrn3524 (2013).
- 61 Nguyen, D., Alushaj, E., Erb, S. & Ito, R. Dissociative effects of dorsomedial striatum D1 and D2 receptor antagonism in the regulation of anxiety and learned approach-avoidance conflict decision-making. *Neuropharmacology* **146**, 222-230, doi:10.1016/j.neuropharm.2018.11.040 (2019).

Similar changes were done to the parallel section in page 14.

9. Page 20: "C57BL/6J mice and SD rats, markedly differ in their social behavior"

This is too broad. Please narrow down to social approach or social investigation behavior or social motivation. Any other social behaviors are not determined in the current study and thus, no conclusions can be drawn regarding species differences in general social behavior.

Answer: We corrected the text as suggested by the reviewer, throughout the text.

10. Abstract: The authors indicate that "mice and rats are used to model human social behavior". I think I know what the authors mean, but this wording less likely represents that. This reads as if rodents display forms of human social behavior. The authors probably want to convey that rodents are used as model organisms to better understand social behavior and its neural underpinnings, which in turn can be informative to understand forms of human social behavior.

Answer: We thank the reviewer for clarifying this issue. The abstract was rephrased as proposed by the reviewer.

11. Abstract: I think that "Equivalently valid as animal models" is the incorrect phrasing. The

question here is whether it matters if you use a specific rat or mouse strain as model organism.

Answer: Abstract was rephrased as justly proposed by the reviewer.

12. I propose to add the following to the last sentence of the abstract: which should be taken into consideration when selecting rats or mice as model organism.

Answer: We thank the reviewer for this helping suggestion. The abstract was rephrased as proposed by the reviewer.

13. Results, page 5: Using the center/periphery location may be a very weak, if at all, measure of stress, unless the authors can find literature suggesting so. In my experience, this is rather a measure of risk assessment and perhaps anxiety-related behavior, but that does not equal stress. It is also unclear how this non-social measure relates to social preference and interactions.

Answer: This point of the reviewer is correct and helpful. We changed the text of this paragraph in the revised manuscript and replaced the term stress with anxiety, which indeed is more appropriate in this context, as suggested by the reviewer. We also added a reference to a recent paper (Harikrishnan et al. 2018), comparing anxiety levels between SD rats and C57BL/6J mice, that reported similar results to ours using both open-field and elevated plus maze tests.

REVIEWER COMMENTS

Reviewer #1 (Remarks to the Author):

The slightly modified manuscript, unfortunately, did not address the reviewer's comments in the last round. The 2-way ANOVA is in fact often works sufficiently well even if the data does not follow normal distribution. The non-parametric test is typically less sensitive than parametric counterparts. If the authors do not find a significant difference between mice and rats with two way ANOVA, it is highly unlikely that the non-parametric test will reveal a difference.

As the manuscript currently stands, the paper compares the social investigation of one specific inbred mouse strain and one specific outbred rat strain and shows that they are different. This difference appears to be unique as it could not be generalized to other strains of the species. As such, it feels like a comparison between green apple and yellow lemon, with a conclusion that they differ in skin color. The study will be more interpretable and a better contribution to the field if the comparison is made between "green apple" and "yellow apple".

Pont-by-point response to the reviewers' comments

Reviewer comments:

The slightly modified manuscript, unfortunately, did not address the reviewer's comments in the last round. The 2-way ANOVA is in fact often works sufficiently well even if the data does not follow normal distribution. The non-parametric test is typically less sensitive than parametric counterparts. If the authors do not find a significant difference between mice and rats with two way ANOVA, it is highly unlikely that the non-parametric test will reveal a difference.

As the manuscript currently stands, the paper compares the social investigation of one specific inbred mouse strain and one specific outbred rat strain and shows that they are different. This difference appears to be unique as it could not be generalized to other strains of the species. As such, it feels like a comparison between green apple and yellow lemon, with a conclusion that they differ in skin color. The study will be more interpretable and a better contribution to the field if the comparison is made between "green apple" and "yellow apple".

Response: We added to the revised manuscript the results of SNP test (on top of the existing SP test results) conducted with two other strains of laboratory rats and mice (both outbred), ICR mice and Wistar Hannover rats (Supp. Fig. 4, last paragraph of page 7). These results show that in the SNP test ICR mice behave like SD rats while Wistar Hannover rats behave similarly to C57BL/6J mice. This made it clear that the behavioral differences observed by us are not species-specific but rather strain-specific, as one mouse strain may behave like one rat strain and vice versa. These results also show that the behavioral differences does not depend on the question whether the strain is inbred or outbred, since the inbred C57BL/6J mice behave in this test like the outbred Wistar rats.

Moreover, we added the data of SP test conducted by another inbred mouse strain, BALB/c mice, which show behavioral pattern which is almost identical to the outbred ICR mice, but significantly distinct from the inbred C57BL/6J mice. These results also show that the question of inbred or outbred strain does not play a role here.

Given these results and the results already presented in the paper for the SP and SNP tests, we have now a complete set of data for five distinct strains (2 rat strains and 3 mouse strains, the latter are one outbred and 2 inbred) and these data are all in agreement with a spectrum of strain-specific differences, where C57BL/6J mice and SD rats presenting two opposite extremes. Accordingly, we screened again throughout the text and made sure that there is no place we make a claim that may sound as if the differences we show are species-specific. We also noted the strain used in any appropriate place and added to the discussion a specific paragraph (2nd paragraph of page 17) discussing why our results demonstrate strain-specific rather than species-specific differences.

As for the statistical test, we changed the test used by us for analyzing these data to Welch's ANOVA, followed by Games-Howell post hoc test. The results of the new set of statistical tests were very similar to those previously obtained by us using Kruskal-Wallis and Mann-Whitney tests, supporting again a species-specific difference.

We believe that these changes address the issues raised by the respected reviewer.

****REVIEWERS' COMMENTS:**

Reviewer #1 (Remarks to the Author):

As the authors have made the effort to state clearly and accurately that this is a strain difference instead of a species difference, the reviewer will now support its publication in Nature Communication.

****REVIEWERS' COMMENTS:**

Reviewer #1 (Remarks to the Author):

As the authors have made the effort to state clearly and accurately that this is a strain difference instead of a species difference, the reviewer will now support its publication in Nature Communication.

Thanks